# Ultra-high thermal stability of sputtering reconstructed Cu-based catalysts

Jiafeng Yu[1], Xingtao Sun[1,2], Xin Tong[1,2], Jixin Zhang[1], Jie Li[3], Shiyan Li[1,2], Yuefeng Liu [1✉], Noritatsu Tsubaki [4✉], Takayuki Abe[5] & Jian Sun [1✉]

The rational design of high-temperature endurable Cu-based catalysts is a long-sought goal since they are suffering from significant sintering. Establishing a barrier on the metal surface by the classical strong metal-support interaction (SMSI) is supposed to be an efficient way for immobilizing nanoparticles. However, Cu particles were regarded as impossible to form classical SMSI before irreversible sintering. Herein, we fabricate the SMSI between sputtering reconstructed Cu and flame-made $LaTiO_2$ support at a mild reduction temperature, exhibiting an ultra-stable performance for more than 500 h at 600 °C. The sintering of Cu nanoparticles is effectively suppressed even at as high as 800 °C. The critical factors to success are reconstructing the electronic structure of Cu atoms in parallel with enhancing the support reducibility, which makes them adjustable by sputtering power or decorated supports. This strategy will extremely broaden the applications of Cu-based catalysts at more severe conditions and shed light on establishing SMSI on other metals.

---

[1] Dalian National Laboratory for Clean Energy, Dalian Institute of Chemical Physics, Chinese Academy of Sciences, 116023 Dalian, China. [2] University of Chinese Academy of Sciences, 100049 Beijing, China. [3] School of Chemistry and Chemical Engineering, Yangzhou University, 225002 Yangzhou, China. [4] Department of Applied Chemistry, School of Engineering, University of Toyama, Gofuku 3190, Toyama 930-8555, Japan. [5] Hydrogen Isotope Research Center, University of Toyama, Gofuku 3190, Toyama 930-8555, Japan. ✉email: yuefeng.liu@dicp.ac.cn; tsubaki@eng.u-toyama.ac.jp; sunj@dicp.ac.cn

Cu-based catalysts have shown large reaction diversity and great potential in many industrial processes[1], such as $NO_x$ removal in automobile applications, steam reforming, CO/$CO_2$ hydrogenation, photo/electro-catalysis, and oxygen reduction reaction. However, their applications are seriously limited by the lack of sufficient thermal stability due to the nature of copper's low Tammann temperature[2], resulting in nanoparticles sintering via surface migration. This process will be further accelerated in the presence of water[2]. As reported, the traditional Cu–Zn–Al catalyst lost 70% activity in 15 h in reverse water gas shift (RWGS) reaction at 600 °C[3]. As a result, the operating temperatures for those catalysts must be restricted, usually to below 300 °C[4,5], inspiring the rational design of anti-sintering Cu-based catalysts at higher temperatures.

It is widely accepted that the encapsulation of metal nanoparticles by reducible oxide overlayer is an efficient way to prevent metal sintering, which constructs physical barriers to separate metal nanoparticles from each other or anchor them from moving[6,7]. This encapsulation phenomenon accompanied with dramatic and reversible effects on small molecular chemisorption was labeled as classical strong metal-support interaction (SMSI) by Tauster et al.[8,9]. It is a specific category with distinctive features, including metal re-dispersion at high reduction temperatures, reversibility in encapsulation, access blocking of molecules, and electron transfer under various redox conditions[10]. Apparently, these features made it distinguish from those general strong interactions between metal and support[11,12]. In general, a high-temperature reduction (about 500 °C) is necessary for this process to create a sub-stoichiometric state with lower oxygen concentrations on reducible oxide supports, and induce oxides migration to the surface of metal nanoparticles[13,14]. Another determining factor of classical SMSI formation is a metal work function. Fu et al.[15] reported that metals with a work function higher than 5.3 eV and a surface energy larger than 2 J m$^{-2}$ could be encapsulated by a titania overlayer, given that the high work function benefited charge transfer from supports to metals[16]. Unfortunately, the Group IB metals including Cu, Ag, and Au were believed very hard to be encapsulated due to their small work functions or low surface energies. Recently, some progress has been achieved on Au nanoparticles at severe conditions (500–700 °C)[10,17–22]. However, the formation of classical SMSI is much harder on Cu particles[7] due to their lower work function (4.46 eV[23]) compared to Au (5.2 eV[24]), as well as weaker ability in $H_2$ activation and dissociation than that of noble metals (Pt, Rh, Pd et al.)[8]. The classical SMSI between Cu and $TiO_2$ support can only be established at an extremely high temperature (close to 1000 °C)[17], where sintering inevitably happened before encapsulation. It remains an enormous challenge to establish classical SMSI at a mild condition and achieve high stability on Cu-based catalysts.

Herein, we designed an ultra-stable Cu-based catalyst for high-temperature RWGS reaction, which consisted of sputtering reconstructed Cu as the active metal and flame-made La-doped $TiO_2$ compound as the support. For metals, the electronic structure of Cu nanoparticles was reconstructed by the bombardment of high-energy plasma in sputtering (SP) technique[25], which promoted the electronic transfer from support to metal and induced the migration of $TiO_x$ species. For supports, the lattice distortion was strengthened during the quenching from the extremely high-temperature process in flame spray pyrolysis (FSP) method[26], which could remarkably enhance the activity of lattice oxygen and reducibility of $TiO_2$, and thus compensate the low capability of Cu in dissociating $H_2$. As a result, a classical SMSI between Cu nanoparticles and $LaTiO_2$ support at low reduction temperatures (300–500 °C) was successfully fabricated. The formation of classical SMSI was identified by reversible

properties in encapsulation, adsorption, and electronic transfer during reduction and oxidation (redox) pretreatments. The present study provided a feasible strategy to establish an ultra-stable Cu-based catalyst at an ultimate temperature as high as 800 °C in a long-time investigation, which opened a new territory in the application of Cu in heterogeneous catalysis.

## Results

**Encapsulation observation.** We constructed the impregnated and sputtered Cu on commercial $TiO_2$ (IM- and SP-Cu/$TiO_2$), in comparison with sputtered Cu on flame-made $LaTiO_2$ (SP-Cu/$LaTiO_2$). After reduction at 500 °C (denoted as -500R), Cu nanoparticles in IM-Cu/$TiO_2$-500R were well dispersed and showed a clean surface (Fig. 1a) with an average particle size of 2.3 nm (Supplementary Fig. 1a). In comparison, a few amorphous species could be detected at the Cu-$TiO_2$ interface in SP-Cu/$TiO_2$-500R, as marked by the arrow in Fig. 1b, showing partial encapsulation formed on sputtered Cu particles. But reduction at 500 °C is not enough to produce sufficient $TiO_x$ for more migration due to the poor reducibility of commercial $TiO_2$. Given this, a 3.8% La-doped $TiO_2$ compound made by FSP ($LaTiO_2$) was used as the support instead of commercial $TiO_2$. Interestingly, for SP-Cu/$LaTiO_2$-500R, Cu particles could hardly be distinguished from support in TEM images (Fig. 1c and Supplementary Fig. 2) but were clearly observed from the annular dark-field (ADF) STEM images with an average particle size around 2.3 nm and EDS elemental mapping (Supplementary Fig. 3), confirming the existence of Cu particles. Usually, metal particles were identified by different contrast from other components or lattice fringes of a specific crystal plane. The missing of visible Cu particles caused by the poor contrast due to the similar atomic number between Cu and Ti ($Z_{Cu} = 29$ vs. $Z_{Ti} = 22$) can be excluded because they can be clearly observed on SP-Cu/$TiO_2$-500R in Supplementary Fig. 1b. The preparation method and the pretreatment process of SP-Cu/$LaTiO_2$-500R and SP-Cu/$TiO_2$-500R were completely the same, except for the doping of 5% La atoms. The effect of La with a much larger atomic number ($Z_{La} = 57$) in the support on the poor contrast can also be excluded since Cu particles appeared after oxidation treatment as seen in Supplementary Fig. 1c for SP-Cu/$LaTiO_2$-ROR sample. These phenomena revealed that the distinction of Cu and $LaTiO_2$ support was practicable, while the indistinguishable species were only Ti oxides layer and $LaTiO_2$ support in this system. The invisible Cu particles were caused by the reduction of Cu contrast via the coverage of the Ti oxides layer on their surface, which has a great effect on TEM rather than STEM. Only a few large Cu particles around 10 nm with an average 1 nm thickness overlayer can be observed in the inset of Fig. 1c. These particles can be identified to be metallic Cu by the lattice fringes of Cu (111) and Cu (200) in Fig. 1e. Moreover, the dynamic encapsulation process with exposure time was discovered upon SP-Cu/$LaTiO_2$-500R from Supplementary Fig. 4 due to the inducement on structural reorganization[27], revealing the migration of support to the Cu surface. The composite of encapsulation was further analyzed by electron energy loss spectroscopy (EELS). Two chemically distinct Ti species in the region I and II were recognized in Fig. 1f, compared to the vacuum area (region III) as a background reference. The splitting of typical $Ti^{4+}$ species with $L_3$ and $L_2$ edges into $t_{2g}$ and $e_g$ levels were observed in the support (region I)[28]. Meanwhile, $TiO_x$ ($x < 2$) species can be identified on the surface of metals (region II) due to a chemical shift towards lower energy loss which resulted from the increase of Ti/O ratio in the dislocation core[29]. After ROR treatment, the appearance of Cu nanoparticles with a clean surface and edges in Fig. 1g was attributed to the removal of encapsulation by oxidation. Likewise,

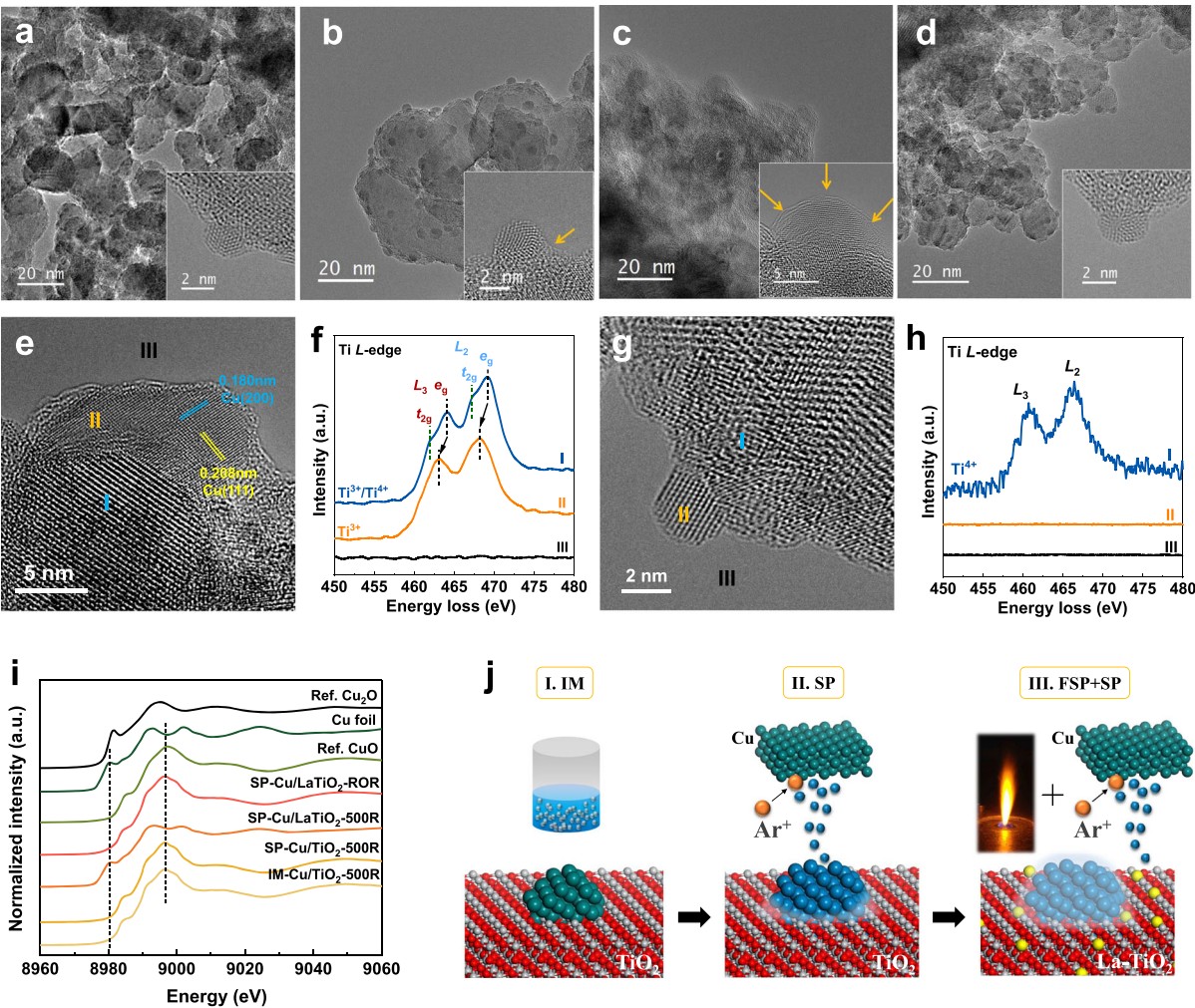

**Fig. 1 Identification and reversibility of encapsulation. a–d** Aberration-corrected HRTEM images of **a** IM-Cu/TiO$_2$-500R, **b** SP-Cu/TiO$_2$-500R, **c** SP-Cu/LaTiO$_2$-500R, and **d** SP-Cu/LaTiO$_2$-ROR (500R-400O-250R). Insets are the magnification of a single Cu nanoparticle, where the orange arrows highlight the oxide overlayer. **e–h** Aberration-corrected HRTEM images (**e**, **g**) and EELS spectra (**f**, **h**) over SP-Cu/LaTiO$_2$-500R (**e**, **f**) and SP-Cu/LaTiO$_2$-ROR (**g**, **h**) catalysts. **i** Cu K-edge X-ray absorption near edge structure (XANES) of different samples, Cu foil, reference Cu$_2$O and reference CuO. **j** Schematic of the development progress of the Cu/TiO$_2$ catalysts in promoting encapsulation, where green, blue, and yellow balls represent normal Cu atoms, sputtered Cu atoms, and La atoms, respectively.

Ti species was no longer detected on the Cu region by EELS (Fig. 1h, region II), suggesting that the encapsulation derived from SMSI was able to be controlled via redox treatments at moderate temperatures. The migration of support to metal particles and exposed clean surfaces again after oxidation treatment was supposed to be one of the main characterizations of classical SMSI as observed on noble metals[17,30]. The encapsulation was also identified by linear EDS analysis in Supplementary Fig. 5. The concentration of Ti, O, and La elements gradually decreased along Cu surface, showing that the Cu nanoparticle was encapsulated by TiO$_x$ species as the profile described in the inset image. Another proof will be given by X-ray absorption spectroscopy (XAS) spectra. Copper species after reduction at 500 °C were studied by Cu K X-ray absorption near edge structure (XANES) spectroscopy via comparison with the data obtained from the reference materials in Fig. 1i. Cu components were analyzed by linear combination fitting (LCF) method in Supplementary Fig. 6 and listed in Supplementary Table 1. Fourier transforms of $k^3$-weighted Cu K EXAFS spectra and their fitting results were shown in Supplementary Fig. 7 and Supplementary Table 2, respectively. Normally, metallic Cu is not stable and can be easily oxidized into Cu$^+$ or Cu$^{2+}$ species in the air. This phenomenon

was observed on both IM-Cu/TiO$_2$-500R and SP-Cu/TiO$_2$-500R samples. No metallic Cu can be detected after exposure in the air, instead, showing typical Cu–O and Cu–Cu bonds in CuO. However, in SP-Cu/LaTiO$_2$-500R, about 70% of Cu still stayed in the metallic state, exhibiting a strong Cu–Cu bond in Cu metal. It indicated that most Cu particles were covered by an oxide layer, considering that metallic Cu was impossible to exist in the air without a protective layer. In comparison, Cu particles were oxidized in the air when encapsulation was removed during oxidation and low-temperature reduction treatments for the same catalyst (SP-Cu/LaTiO$_2$-ROR).

The classical SMSI was hardly occurred on the traditional IM-Cu/TiO$_2$ catalyst due to the lower work function nature of Cu and its weaker capability in promoting Ti$^{4+}$ reduction compared to noble metals[15]. We showed the schematic of the development progress of Cu-based catalysts in promoting encapsulation in Fig. 1g. The first step was from (I) IM-Cu/TiO$_2$ to (II) SP-Cu/TiO$_2$ by reconstructing metallic Cu with the SP method to lower the energy barrier of electron transfer. According to our previous study[25], the electronic structure of Cu atoms can be changed under the high-energy argon plasma bombardment during the SP process. The outermost electrons tended to approach interlayers

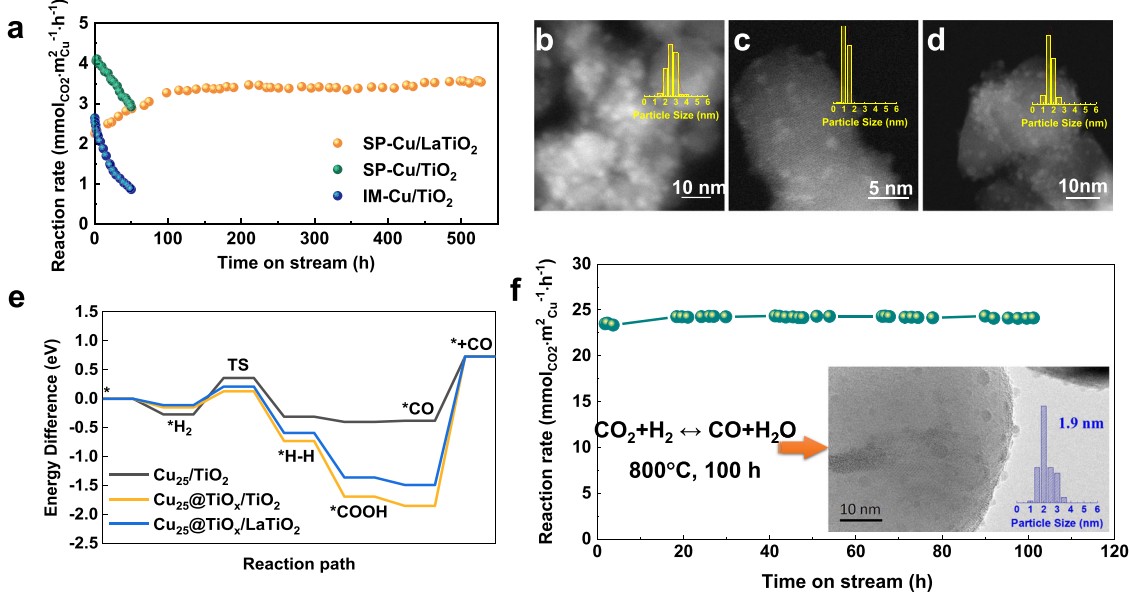

**Fig. 2 Thermal stability of Cu-based catalysts at high temperatures. a** Stability tests in RWGS reaction at 600 °C over IM-Cu/TiO$_2$, SP-Cu/TiO$_2$, and SP-Cu/LaTiO$_2$ samples. **b–d** ADF-STEM images and particle size distributions of **b** SP-Cu/LaTiO$_2$-500R, **c** SP-Cu/LaTiO$_2$-100h, and **d** SP-Cu/LaTiO$_2$-500h catalysts. **e** Comparison of the RWGS reaction path over three models from DFT calculations. **f** Long-term thermal stability of SP-Cu/LaTiO$_2$ at 800 °C. Inset was the TEM image and the distribution of Cu nanoparticles of the used catalyst.

via the enhanced "penetration effect", resulting in the increment of the energy for the outmost electron to escape from atoms (i.e., the work function). Although the classical SMSI became possible on SP reconstructed Cu, it was still restricted by Ti$^{4+}$ reduction. The second step was from (II) SP-Cu/TiO$_2$ to (III) SP-Cu/LaTiO$_2$ by regulating TiO$_2$ support with the FSP method. According to XPS results in Supplementary Fig. 8, a splitting of the La $3d_{5/2}$ and La $3d_{3/2}$ lines was observed in the La $3d$ core-level spectra. For La $3d_{3/2}$, the peaks at 851.7 and 855.4 eV were identified as the main and the shake-up satellite peaks, respectively. The binding energy difference between the main and satellite peaks ($\Delta E$) was 3.7 eV. Accordingly, $\Delta E$ should be the same for La $3d_{5/2}$, i.e., the main and the shake-up satellite peaks should be at 834.8 and 838.5 eV, respectively. In addition to the peak position and splitting, $\Delta E$ in the multiples split can be used to distinguish La$_2$O$_3$ from other La$^{3+}$ compounds. Usually, the $\Delta E$ values for La$^{3+}$ compounds are in the range of 3.5 to 4.6 eV for the La $3d_{5/2}$ spectrum, where the one for La$_2$O$_3$ was 4.6 eV. Therefore, this La XPS profile revealed the presence of a mixture of lanthanum oxide species. Furthermore, we found that the one at 834.8 eV disappeared and split into two peaks, which can be assigned to Ti–O–La species (834.0 eV[31]) and La$_2$O$_3$ (835.4 eV[32]), indicating that Ti–O–La species were co-existing with pure La$_2$O$_3$ in SP-Cu/LaTiO$_2$ sample. The intense peak of O $1s$ at lower binding energy 529.7 eV was assigned to the lattice oxygen in TiO$_2$ (Ti–O bonds). The peak at 531.0 eV may belong to oxygen atoms in the surface hydroxyl groups (H–O bonds)[33]. An additional peak at 532.2 eV can be attributed to Ti–O–La[34,35]. Typical peaks of Ti $2p_{3/2}$ and Ti $2p_{1/2}$ could be detected at 458.3 and 464.2 eV, respectively. The splitting between the two core levels was close to TiO$_2$ reference (5.7 eV), indicating a valence state of +4 for Ti in the support. Compared to SP-Cu/TiO$_2$ sample, extra peaks at 459.5 and 465.2 eV with an energy difference of 5.7 eV were found, which was probably related to Ti–O–La species, because it has been reported that the binding energy of Ti $2p_{3/2}$ shifted to higher energy when adding La$^{3+}$ species[36]. From the analysis above, La atoms substituted Ti atoms in the form of Ti–O–La, inducing lattice disorder by the radius disparity between La$^{3+}$ (1.03 Å) and

Ti$^{4+}$ (0.61 Å)[32]. A small amount of La$_2$O$_3$ was observed from XRD (Supplementary Fig. 9), indicating that 3.8 wt.% La was excessive for doping into TiO$_2$ lattice. The ratio of anatase and rutile calculated by the XRD pattern was 82%, which was consistent with the information from the reagent (80%). The ratio in LaTiO$_2$ support was 62%, which may be caused by the high-temperature process in the FSP method. From the ADF-STEM images in Supplementary Fig. 10, the bright particles marked by red circles in SP-Cu/LaTiO$_2$ sample with a size about 2 nm can be assigned to Cu nanoparticles, while the bright dots marked by yellow circles can be identified as single La atoms in the form of La–O–Ti structure compared to SP-Cu/TiO$_2$, indicating La was atomically dispersed in TiO$_2$ lattice. A high degree of non-stoichiometry active oxygen was created during the high-temperature quenching process in the FSP method[26], showing better reducibility than commercial TiO$_2$ according to TPR results in Supplementary Fig. 11.

**Catalytic performance**. The reverse water gas shift reaction (RWGS, CO$_2$ + H$_2$ = CO + H$_2$O) is a pivotal process in CO$_2$ utilization and further catalytic conversion to liquid fuels[37]. Long-time stability of catalysts over RWGS at 600 °C was evaluated to reveal the function of encapsulation on immobilizing Cu particles, especially in the presence of water. Considering the variation of exposed active sites with coverage extent, the exposed Cu surface area of tested catalysts was measured and listed accompanied with Cu loading and dispersion in Supplementary Table 3. From Fig. 2a, the reaction rate over IM-Cu/TiO$_2$ obviously decreased with a decline rate of 65% in 50 h due to aggregation and sintering as shown in Supplementary Fig. 12. After the reaction, the average particle size of Cu increased from 2.3 to 4.7 nm with a much broader distribution, indicating that severe thermal sintering happened on Cu particles during the reaction. In comparison, SP-Cu/TiO$_2$ showed not only a higher initial reaction rate but also a slower decline rate (29% in 50 h). The designed SP-Cu/LaTiO$_2$ catalyst further exhibited unexpectedly excellent thermal stability without any sign of deactivation during

a long time investigation (more than 500 h) even in the presence of water at 600 °C, which was a breakthrough for Cu-based catalysts[38] compared with the traditional Cu-Zn-Al catalyst (70% activity lost in 15 h in RWGS at 600 °C[3]). During the reaction, the initial rate gradually increased from 2.3 to 3.5 $mmol_{CO2}$ $m^2_{Cu}{}^{-1}$ $h^{-1}$ and kept constant for about 100 h. On one hand, Cu nanoparticles were re-dispersed by SMSI as reported[21,39,40] with the average particle size decreasing from 2.3 to 0.9 nm in the first 100 h and gradually increasing afterward until reaching stable to 1.5 nm after 500 h reaction as observed in Fig. 2b–d. Cu dispersion increased from 13.5 to 37.6% according to $N_2O + TPR$ measurements (Supplementary Fig. 13). The re-dispersion phenomenon of metal nanoparticles induced by the effect of SMSI was widespread[39,40]. During the high-temperature reduction process, a large number of oxygen vacancies were produced on the reducible support. Meanwhile, the migration of Cu atoms from the Cu nanoparticles to the support will happen during the thermal motion process. These Cu atoms will be captured and settled by the oxygen vacancies under the strong metal-support interaction, thus inducing the re-dispersion of metal particles. According to the evolution of particles size, the Ostwald ripening process and re-dispersion phenomenon might co-exist in the whole reaction process in a restrictive relation. On the other hand, the extent of encapsulation and the exposure of Cu active sites may be in a dynamic equilibrium process during reaction in the combined effects of $H_2$ reduction and $H_2O$ oxidation. It has been reported that the overlayer derived from the SMSI between Rh and $TiO_2$ can be oxidized in the humid environment of the $CO_2 + H_2$ reaction[41]. NO reduction by CO to produce $N_2$ and $CO_2$ was performed as a probe reaction after $H_2O$ and $H_2$ treatments to evaluate the variation of coverage as shown in Supplementary Fig. 14. After pretreatment in $H_2$ (600H), Cu nanoparticles were fully covered by $TiO_x$ species, leading to complete deactivation. Then, the sample was pretreated in a 15%

$H_2O/He$ atmosphere at 600 °C for 1 h. CO conversion increased to 71.37% but was still lower than the initial conversion (89.13%), indicating that the coverage was partially removed by $H_2O$ oxidation. Afterward, the sample was further pretreated in a 15% $H_2O/H_2$ atmosphere at 600 °C for 1 h. CO conversion decreased to 39.04%, demonstrating that the removed coverage can be partially recovered by the addition of $H_2$ in the $H_2O$ atmosphere. The combined reduction and oxidation effects in the system went through a dynamic equilibrium process until the coverage extent got a stable state. Accordingly, the exposed Cu surface area decreased to 12.7 $m^2 g^{-1}$ after reduction and recovered to 35.4 $m^2 g^{-1}$ after the reaction. When the reaction stopped, the entire encapsulation formed again in the $H_2$ protection atmosphere as observed in Supplementary Figs. 15 and 16.

The molecule-level mechanism over the above Cu-based catalysts in RWGS reaction was conducted by DFT calculations as displayed in Fig. 2e, Supplementary Fig. 17 and Supplementary Table 4. The RWGS reaction path is as follows. Firstly, hydrogen is adsorbed on the catalyst, followed by the dissociation of $H_2$ to H atoms on Cu sites. Subsequently, an active hydrogen atom adsorbs $CO_2$ to form *COOH intermediate, and another hydrogen atom moves to *COOH thus producing $H_2O$. The last step is the desorption of CO. From Fig. 2e, the activation energy barrier of hydrogen dissociation to form a transient state (TS) on $Cu_{25}/TiO_2$ was 0.63 eV, rather higher than that of $Cu_{25}@TiO_x/TiO_2$ (0.28 eV). It demonstrated a more facile activation of hydrogen to active H atom, probably resulting from partially encapsulated $TiO_x$ surface. La doping into $TiO_2$ slightly increased the barrier to 0.32 eV. Accordingly, the SP made Cu series catalysts with an encapsulation followed a more favorable process with multiple steps in the high-temperature RWGS reaction compared to the conventional catalyst. The turnover frequency (TOF) of SP-Cu/$TiO_2$ measured under kinetic conditions at 600 °C was 0.11 $s^{-1}$, which was almost five times more than that

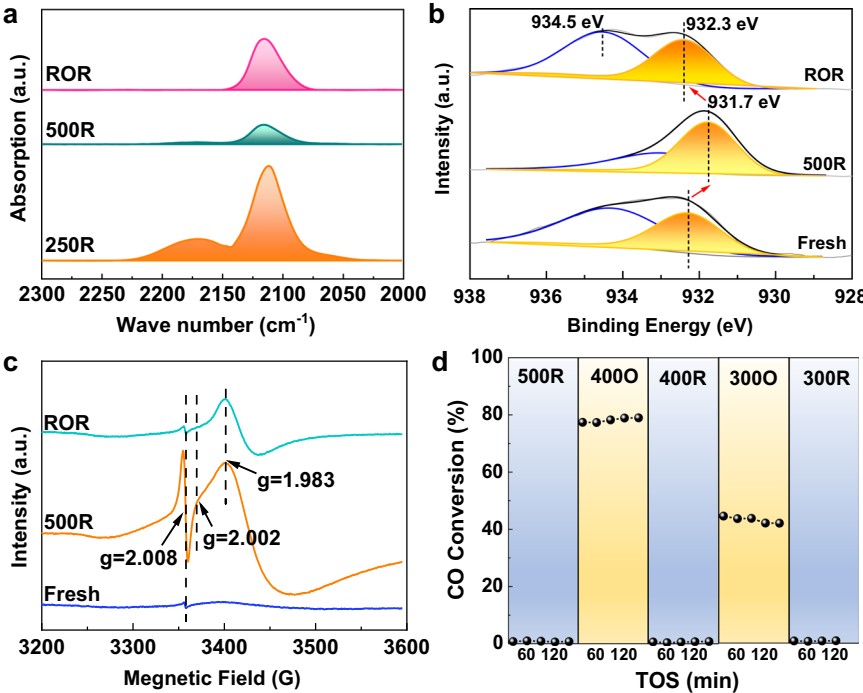

**Fig. 3 Identification of classical SMSI over SP-Cu/LaTiO₂ catalyst. a** CO-IR spectra of fresh SP-Cu/LaTiO₂ sample with a successive reduction and oxidation treatments: reduced at 250 °C (-250R), further reduced at 500 °C (-500R), then oxidized at 400 °C and again reduced at 250 °C (-ROR). **b**, **c** Quasi in-situ XPS of Cu 2$p$ (**b**) and EPR spectra (**c**) of SP-Cu/LaTiO₂-fresh, SP-Cu/LaTiO₂-500R and SP-Cu/LaTiO₂-ROR samples. **d** Reversibility of catalytic performance over CO + NO reaction after redox pretreatments. CO conversions in CO + NO reaction were collected at 250 °C. Before each reaction cycle, samples were pretreated with 99.99% H₂ (XR) or 5% O₂/He (XO) at different temperatures for 1 h, where X represents the temperatures.

of IM-Cu/TiO$_2$ (Supplementary Table 3). The apparent activation energy ($E_a$, shown in Supplementary Fig. 18) of SP made Cu/TiO$_2$ (68 ± 4 kJ mol$^{-1}$) was much lower than the IM made one (82 ± 4 kJ mol$^{-1}$) and typical Cu–Zn–Al catalysts (87 kJ mol$^{-1}$) in literature[42], owing to the low activation energy barrier of hydrogen dissociation to form a transient state (TS) on the Cu sites of SP made catalysts.

Afterwards, the outstanding stability of SP-Cu/LaTiO$_2$ was further confirmed by a high-temperature RWGS test at 800 °C in Fig. 2f. The average Cu particle size could keep as small as 1.9 nm even went through 100 h reaction in the presence of water, demonstrating that the sintering of Cu nanoparticles could be effectively prevented by SMSI at a temperature as high as 800 °C, which is much more stable than any of Cu-based catalysts reported so far.

**Classical SMSI identifications.** The classical SMSI is well known for its distinctive features of reversibility in blocking the access of molecules, electron transfer, Ti valence states and encapsulation under reduction-oxidation treatments. FT-IR was used to investigate the variation of exposed Cu surface during redox treatments via CO adsorption. In Fig. 3a, the bands of CO adsorbed on exposed metallic Cu (2112 cm$^{-1}$) and Cu oxides[43] (2177 cm$^{-1}$) co-existed after low-temperature reduction (250R). Apparently, this temperature was not enough to completely reduce Cu oxides, but further increasing temperature to 300 °C would cause encapsulation (Supplementary Fig. 19). Reduction at a higher temperature will induce further encapsulation (500R). After redox treatments (ROR), reversible CO adsorption was observed. A similar phenomenon was also found for SP-Cu/TiO$_2$ but not for IM-Cu/TiO$_2$ as shown in Supplementary Fig. 20. Quasi in situ XPS over SP-Cu/LaTiO$_2$ sample was conducted after various treatments as shown in Fig. 3b. Details of Cu valence state analysis were shown in Supplementary Fig. 21. The Cu 2$p_{3/2}$ of Cu$^0$ and Cu$^{2+}$ species are at the binding energy of 932.3 and 934.5 eV, respectively. Cu 2$p_{1/2}$ is identified according to split spin-orbit components of Cu 2$p$ peak with the fixed splitting of the doublet $\Delta = 19.75$ eV and intensity ratio at 0.508. The XPS result demonstrated that the surface of sputtered Cu in the SP-Cu/LaTiO$_2$-fresh sample was oxidized due to the passivation process during the SP method. For Cu 2$p$, the peak at 932.3 eV shifted to low energy by 0.6 eV after reduction, while the peak at 458.3 eV for Ti 2$p$ shifted to high energy by 0.5 eV (Supplementary Fig. 22). Then they shifted back after the subsequent OR treatment, confirming that electrons reversibly transferred from support to metal. Ex-situ XAS measurements were conducted to evaluate the formation of encapsulation layer via their protective effect on metallic Cu from being oxidized in the air for different catalysts (Supplementary Figs. 23, 24 and Supplementary Table 5). For fresh SP-Cu/LaTiO$_2$ sample, Cu was identified to be a metallic state, because the electronic structure was reconstructed by Ar plasma sputtering, making Cu atoms hardly be oxidized in the air[25]. This unique electronic structure will not be destroyed even after reduction at 250 °C. Furthermore, Cu was still stable in the metallic state, no matter it was reduced at high temperature or exposed in reaction atmosphere, as shown in SP-Cu/LaTiO$_2$-500R and -100h catalysts, respectively, exhibiting the encapsulation on the surface of Cu could efficiently protect it from being oxidized, making Cu metallic state keep unchanged when exposing to the air. Electron paramagnetic resonance (EPR) measurements (Fig. 3c) showed that the SP-Cu/LaTiO$_2$ catalyst underwent a reversible transformation of oxygen vacancies ($g = 2.008$), superoxide O$_2^-$ (hole traps, $g = 2.002$), and Ti$^{3+}$ sites (electron traps, $g = 1.983$) during redox treatments[44,45], which was not observed in those samples supported by commercial TiO$_2$

(Supplementary Fig. 25). Furthermore, NO + CO reaction was conducted at 250 °C after each redox pretreatment since it was very sensitive to the encapsulation variations. From Fig. 3d, all of the reduction between 300 and 500 °C could induce the formation of encapsulation on SP-Cu/LaTiO$_2$, leading to its total deactivation. The recovery of activity once oxidized in O$_2$ revealed the removal of encapsulation, and the recovery degree depended on the oxidation temperatures.

**Classical SMSI regulations.** High extent encapsulation is more effective in stabilizing Cu nanoparticles but will sacrifice more active sites. Therefore, a more sophisticated regulation of SMSI strength was expected to find a balance between activity and stability. The encapsulation extent mainly depends on metal work function and support reducibility. For metal, the Cu work function should have a relationship with the power of high-energy argon plasma bombardment during the SP process by reconstructing electronic structure at different extents. As shown from Cu LMM spectra of X-ray excited Auger electron spectroscopy (XAES) in Fig. 4a, Cu electron structures of SP-Cu/LaTiO$_2$ catalyst made with four different power were remarkably different, despite showing similar Cu 2$p$ spectra (Supplementary Fig. 26). The primary sharp feature at the kinetic energy of 918.6–918.9 eV and 921.3–921.7 eV in the spectrum for Cu was attributed to the $^1$G and $^3$F multiplet of the localized 3$d^8$ final state. The low energy peak at ca. 916.7 and 913.7 eV was ascribed to Auger vacancy satellite structure from L$_2$L$_3$M$_{45}$ Coster–Kronig transition, termed as $^4$F* and $^2$F*, respectively[46,47]. With the increasing sputtering power from 100 to 300 W, both of the two primary peaks of Cu ($^1$G and $^3$F) and the secondary peak ($^4$F*) gradually shifted towards higher kinetic energy with a maximal amplification value of 0.4 eV and subsequently maintained unchanged when further increasing the power to a maximal 450 W (Supplementary Table 6). The higher kinetic energy of excited Auger electrons suggested the existence of higher energy electrons in the Cu L$_3$ and M$_{45}$ energy levels, resulting from the high-energy plasma bombarding with a higher sputtering power, providing a new strategy to control the work function of Cu.

The catalytic stability of samples made with different power was tested in RWGS reaction, and the variation of CO$_2$ conversion compared to the initial activity was plotted in Fig. 4b. For SP-Cu/LaTiO$_2$-100W, the CO$_2$ conversion was quickly decreased by almost 60% in 150 h, revealing that the formed encapsulation at this condition was not enough to protect Cu particles from sintering. The deactivation rate greatly declined (10% in 100 h) when increasing power to 200 W. The deactivation was completely avoided at 300 W. Dynamic regulation by reaction atmosphere happened at 450 W due to high extent encapsulation as discussed previously.

As shown in Fig. 4c, complete coverage of Cu particles happened on SP-Cu/LaTiO$_2$ after reduction at a temperature higher than 300 °C, while it increased to higher than 500 °C for ZnZrO$_2$ support. In comparison, only a 20% decrease of CO conversion can be seen after reducing at 500 °C for commercial TiO$_2$ support. The SMSI cannot be constructed even by increasing reduction temperature as high as 700 °C for CeO$_2$ support, but can partially form on CeZrO$_2$ support. Therefore, the encapsulation extent can be also regulated by different supports due to their different reduction behaviors.

In summary, we successfully established an ultra-stable Cu-based catalyst at high temperatures by constructing classical SMSIs. It was demonstrated that the combined strategies of reconstructing the electronic structure of Cu atoms via plasma sputtering and enhancing the oxygen lattice distortion of supports by flame spray pyrolysis were critical for SMSI formation. Typical

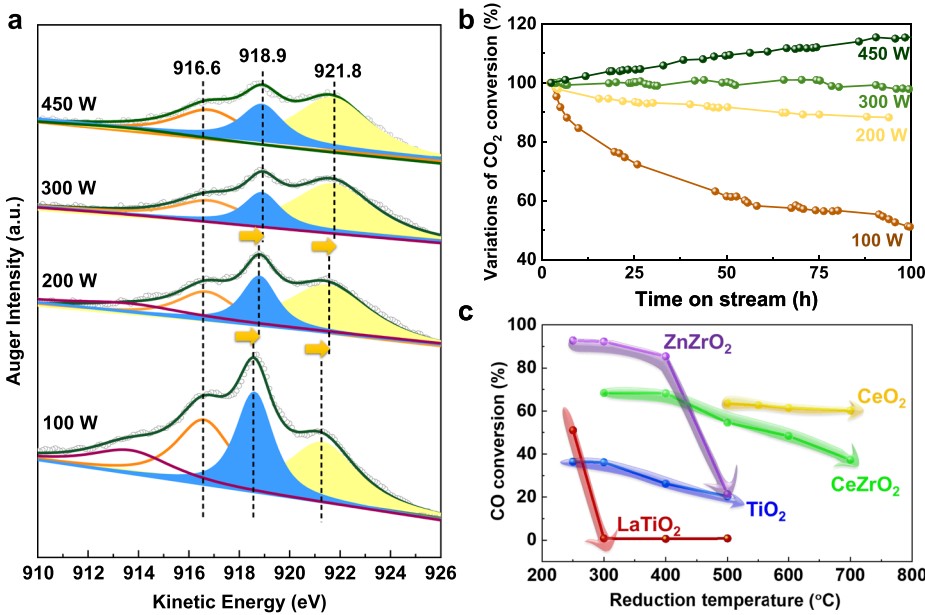

**Fig. 4 Regulation of classical SMSI on Cu-based catalysts. a** Cu LMM XAES spectra of SP-Cu/LaTiO$_2$ with different energy power. **b** CO$_2$ conversion variation with TOS during RWGS reaction at 600 °C. **c** CO conversion of Cu-based catalysts by sputtering Cu on various supports, such as LaTiO$_2$, TiO$_2$, ZnZrO$_2$, CeO$_2$, and CeZrO$_2$ in CO + NO reaction as a function of reduction temperature in the pretreatments, where the reduction was conducted at different temperatures for 1 h in pure H$_2$ before each reaction at 250 °C.

properties of classical SMSI, such as reversible encapsulation, electron transfer, small molecular adsorption and catalytic activity after redox pretreatments were well demonstrated. Extremely stable performance was achieved at 600 °C for more than 500 h, and Cu nanoparticles were effectively confined by SMSI even at as high as 800 °C for SP-Cu/LaTiO$_2$ catalyst. Significantly, the strength of SMSI can be effectively controlled not only by reduction temperatures as before but also by tailor-made Cu via sputtering power regulation or decorated supports for rational catalyst design on demands. These discoveries unlock the strategy of inducing classical SMSI on widespread non-noble metals. We expect that our new findings will serve as a general guide to design a highly stable heterogeneous catalyst at a wider temperature.

## Methods

**Synthesis of LaTiO$_2$.** A La-doped TiO$_2$ supports were prepared by flame spray method (FSP) with a setup described in Supplementary Fig. 27[48]. Briefly, the precursor solution was prepared by dissolving 1.73 g lanthanum acetylacetonate (Aladdin of China) in 210 ml of benzyl alcohol and treated by ultrasound for 1 h. Then 32.78 g tetrabutyl titanate (Aladdin of China) was added to the above mixture with ultrasonic treatment for another 1 h. The homogeneous precursor with a metal concentration of 0.4 mol l$^{-1}$ was obtained. The flame was generated by burning a mixture of CH$_4$ (99.99%, 0.6 l min$^{-1}$) and O$_2$ (99.99%, 1.9 l min$^{-1}$). Additional dispersion gas (99.99% O$_2$, 3.5 l min$^{-1}$) was fed through the gap around the precursor solution nozzle at a pressure of 1.5 bar. Abundant air (5 l min$^{-1}$) was provided through a sintered metal plate ring to stabilize the flame. The precursor solution was supplied to the flame at a flow rate of 5 ml min$^{-1}$ by a syringe pump. The particles were collected on a water-cooled glass fiber filter (WhatmanGF/D, 25.7 cm in diameter) with the aid of a vacuum pump. The collected particles were ready to use without further calcination process. The oxide support was named LaTiO$_2$, where La content was identified to be 3.8 wt.% according to the ICP result. The ZnZrO$_2$, CeO$_2$, CeZrO$_2$ compounds were synthesized by the FSP method following the above steps with atomic ratio of Zn:Zr = 1 and Ce:Zr = 1.

**Synthesis of SP-Cu/LaTiO$_2$ and SP-Cu/TiO$_2$.** The Cu component was deposited by the sputtering (SP) method with a setup described in Supplementary Fig. 28[25]. A pure Cu target (purity > 99.9%; 50 mm by 100 mm; Toshima Co. Ltd.) was firstly pre-sputtered for 0.5 h to make the surface clean. About 3 g LaTiO$_2$ fine powder was put into a hexagonal rotating barrel. The vacuum chamber was filled with Ar (99.995%, 29 ml min$^{-1}$) at a pressure of 2.0 Pa. The hexagonal barrel was rotated at 3.5 rpm and vibrated mechanically to mix the support. The SP was executed with the input power of 450 W for 3 h to deposit Cu nanoparticles on the spinning

support. After that, a 1% O$_2$/N$_2$ flow was gradually introduced into the barrel to recover ordinary pressure and passivated the surface of metallic Cu. The obtained catalyst was denoted as SP-Cu/LaTiO$_2$. The SP-Cu/TiO$_2$ catalyst was prepared following the same procedure with commercial TiO$_2$ (P25, Degussa) as support. For comparison, Cu-based catalysts through sputtering Cu on various FSP-synthesized supports, such as ZnZrO$_2$, CeO$_2$, and CeZrO$_2$, were prepared following the above steps, which are denoted as SP-Cu/ZnZrO$_2$, SP-Cu/CeO$_2$, and SP-Cu/CeZrO$_2$, respectively. Moreover, Cu was sputtered on LaTiO$_2$ with different input power (100, 200, 300, and 450 W) by SP method, named SP-Cu/LaTiO$_2$-100W, -200W, -300W and -450W, respectively. Cu loadings of each sample were 26.7, 19.2, 11.9, and 18.8 wt.%, respectively, according to ICP identification.

**Synthesis of IM-Cu/TiO$_2$.** About 12 wt.% Cu was loaded on a commercial TiO$_2$ (P25, Degussa) support with the impregnation method. The appropriate amount of TiO$_2$ powder was added into the Cu aqueous solution which was made by dissolving 0.532 g CuNO$_3$·3H$_2$O (Kermel) into 1.5 ml of deionized water. After impregnation for 6 h, the catalyst was evaporated at 80 °C for 2 h, dried at 120 °C for 12 h, and then calcined in a muffle oven at 300 °C for 4 h. The obtained catalyst was denoted as IM-Cu/TiO$_2$.

**Pretreatments.** Fresh samples were pretreated under reduction or oxidation conditions. Taking SP-Cu/LaTiO$_2$ as an example, it was subsequently reduced under H$_2$ flow (99.99%, 30 ml min$^{-1}$) for 1 h at different temperatures under atmospheric pressure, and each was denoted as SP-Cu/LaTiO$_2$-XR (X represents the reduction temperature). The SP-Cu/LaTiO$_2$-500R sample was oxidized under O$_2$/He flow (5 vol.%, 30 ml min$^{-1}$) at 400 °C for 1 h and denoted as SP-Cu/LaTiO$_2$-500R-400O (abbreviated as -RO). Then it was reduced again at low temperature (250 °C) to expose the metallic Cu, which was named as SP-Cu/LaTiO$_2$-500R-400O-250R (abbreviated as -ROR). It will be denoted as SP-Cu/LaTiO$_2$-XW or SP-Cu/LaTiO$_2$-XWH (X represents the pretreatment temperature) when it was pretreated in 6% H$_2$O/He (W) or 6% H$_2$O/H$_2$ (WH) atmosphere, respectively. The same denotation rules were also applied in other samples.

**Redox pretreatments.** NO reduced by CO reaction (2NO + 2CO = N$_2$ + 2CO$_2$) was conducted in a fix-bed quartz reactor with an inner diameter of 8 mm. 100 mg sieved catalyst at sieve fraction of 425–850 μm was pretreated in alternative reduction and oxidation cycles. The samples were reduced at 500, 400, and 300 °C for 1 h at atmospheric pressure in a 99.99% H$_2$ flow (50 ml min$^{-1}$) at the 1st, 3rd and 5th cycle and oxidized at 400 and 300 °C for 1 h in a 5 vol.% O$_2$/He flow (50 ml min$^{-1}$) at the 2nd and 4th cycle. The pretreatments were named XR and XO for reduction and oxidation, respectively, where X represents the treatment temperature. After each pretreatment, the sample was purged by He flow (50 ml min$^{-1}$) for 30 min and then cooled down to 250 °C. A feed gas (5 vol.% CO + 5 vol.% NO balanced with He) was introduced into the reactor under atmospheric pressure with a flow rate of 25 ml min$^{-1}$. CO conversion was calculated according to the CO

and $CO_2$ compositions in the outlet gas after 30 min reaction which was analyzed by an online gas chromatograph (Shimadzu GC-14C) equipped with a TDX-01 column.

**Water or hydrogen pretreatments**. A similar experiment was conducted to evaluate the variation of coverage in the pretreatment of water or hydrogen. CO + NO reaction was conducted at 300 °C for SP-Cu/LaTiO$_2$ sample. Then the sample was heated to 600 °C for pretreatment in three different atmospheres, i.e., pure H$_2$ (600H), in 15 vol.%H$_2$O(g)/He (600 W), and 15 vol.%H$_2$O(g)/H$_2$ (600WH). After each pretreatment step, the sample was cooled down to 300 °C and tested in NO + CO reaction.

**RWGS reaction**. The reverse water gas shift (RWGS) reaction ($CO_2 + H_2 = CO + H_2O$) was performed in a fix-bed quartz reactor. 100 mg sieved catalyst (250–380 μm) was firstly reduced at 500 °C for 1 h in a pure hydrogen flow (50 ml min$^{-1}$), then heated up to 600 or 800 °C in He (99.999%, 50 ml min$^{-1}$). Then a mixture of H$_2$ and CO$_2$ with a volume ratio of 2 was passed over the catalysts bed with a total flow rate of 25 ml min$^{-1}$ under atmospheric pressure. The CO and CO$_2$ compositions in the outlet gas were monitored with reaction time by the same online gas chromatograph above. The CO$_2$ conversion at 1 h was regarded as the initial activity. The variation of CO$_2$ conversion was defined as the percentage of the difference between initial activity and current conversion compared to the initial activity. The preparation and the stability tests of both SP-Cu/TiO$_2$ and SP-Cu/LaTiO$_2$ samples were repeated by different people with different setups more than two times. The results showed well repeatability.

**Kinetic analysis**. The apparent activation energy ($E_a$) of catalysts was determined using the same reactor and reduction procedure as catalytic tests at atmospheric pressure. 20 mg samples with 100–150 μm particle size were mixed with 80 mg quartz sand to exclude internal diffusion effects. RWGS reaction was performed at a high GHSV of 40,800 ml g$_{cat}^{-1}$ h to exclude external diffusion effects. The temperature was varied between 500 and 600 °C. The apparent activation energy was calculated according to the Arrhenius equation. The CO$_2$ conversion was calculated according to the CO and CO$_2$ compositions in the outlet gas which were analyzed by an online gas chromatograph (Shimadzu GC-14C) equipped with a TDX-01 column. Water was cooled down before going into GC. No other products were detected. TOF of each catalyst was calculated according to the initial activity in 2 h at 600 °C and the amount of exposed Cu measured by the N$_2$O method.

**Electron microscopy**. High-resolution transmission electron microscopy (TEM) were operated on the Titan Themis G3 ETEM microscope (ThermoFisher) equipped with a spherical-aberration corrector (CEOS GmbH) for the parallel imaging at 300 kV and measured the information resolution better than 1.0 Å. The TEM images were recorded by a Gatan Oneview camera in 4096 × 4096 pixels[2] resolution under an exposure time of 2.0 s at an electron current density of 3.2 A cm$^{-2}$ condition[49]. The electron energy loss spectroscopy (EELS) data were obtained in scanning transmission electron microscopy (STEM) mode equipped with a CCD detector, which is back of an EEL spectrometer (Gatan Imaging Filter, GIF quantum model 965). The energy resolution of EELS is better than 100 meV (determined from full-width at half-maximum (FWHM) of the zero-loss peak). The convergence angle for the experiments was 9.8 mrad and the collection angle for the GIF system was 33–198 mrad. The spectral sampling was 0.25 eV per channel. Atomic resolution annular dark-field (ADF) STEM images were conducted on a Hitachi HF5000 cold-field microscope with a probe Cs-corrector operated at 200 kV. The energy-dispersive X-ray spectroscopy (EDS) elemental mapping and line scanning results were conducted at symmetrically opposed dual 100 mm$^2$ EDX* detectors (Symmetrical Dual SDD*) equipped on the HF5000 microscope.

**CO chemisorption**. The adsorption of CO on Cu surface was studied using a Nicolet 6700 IR spectrometer equipped with a DRIFT (diffuse reflectance infrared Fourier transform) cell. The sample was firstly loaded in the cell and then pretreated in 10%H$_2$/Ar (30 ml min$^{-1}$) at the specific reduction temperature for 0.5 h, followed by Ar (30 ml min$^{-1}$, 99.99%) sweeping for 0.5 h at the same temperature. After cooling down to 30 °C, the background was scanned in an Ar flow. The infrared spectrum of CO adsorption (CO-IR) was successively recorded every 5 min starting from introducing 5% CO/Ar (30 ml min$^{-1}$) into the IR cell for 30 min. Subsequently, CO was switched into Ar to sweep the CO gas until the adsorption state was steady. The spectral resolution was 4 cm$^{-1}$ and the number of scans was 64.

**XPS**. Quasi in-situ X-ray photoelectron spectra (XPS) were recorded on the Thermo Scientific ESCALAB 250Xi spectrometer with a monochromatic Al Kα source radiation and a spot size of 500 μm in diameter. It operated with an analyzer in constant analyzer energy (CAE) mode with pass energy and energy step size of 100 and 1 eV for the survey, respectively. They were 20 and 0.05 eV for high-resolution spectra. Surface charging effects occurring in insulant samples were avoided using an electron flood gun. The XPS measurement was conducted at

room temperature. XPS measurements of C 1s, O 1s, Cu 2p, Cu LM2, Ti 2p, and La 3d binding energies were conducted at room temperature and atmospheric pressure for the fresh sample (Fresh). Then it was reduced at 500 °C for 1 h in 99.99% H$_2$ flow at 0.5 MPa with a heating rate of 10 °C min$^{-1}$. After cooling to room temperature, the samples were inertly transferred to a glove box connected to the spectrometer and mounted on the sample holder. They were directly transferred from the glove box to the spectrometer chamber without air contact. Then XPS spectra were measured in vacuum (500R). Subsequently, the sample was oxidized at 400 °C for 1 h in 10% O$_2$/He flow at 0.1 MPa and cooled down to 250 °C for another reduction process of 1 h in 99.99% H$_2$ flow at 0.5 MPa. The binding energy was calibrated by contamination carbon C 1s peak (284.8 eV) as the reference. After that, XPS spectra were measured again, named ROR. Ex-situ XPS with Auger electron spectroscopy (AES) was also conducted on the same equipment. Before the test, samples were pretreated by Ar ionic SP with a rate of 0.16 nm s$^{-1}$ for 100 s. The collected BEs in XPS were calibrated using the C 1s peak at 284.8 eV as the reference.

**EPR**. Low-temperature electron paramagnetic resonance (EPR) spectra were collected at 100 K by a Bruker A200 EPR Spectrometer. Equal amounts of the catalysts were placed into a homemade quartz tube with stopcocks. The catalysts were pretreated at 500 °C for 1 h in 99.99% H$_2$ flow (500R). Then part of these samples was continuously oxidized at 400 °C for 1 h in 5% O$_2$/He flow followed by reduction at 250 °C for 1 h in 99.99% H$_2$ flow again (ROR).

**TPR**. Hydrogen temperature-programmed reduction (H$_2$-TPR) experiments were performed with a self-made TPR system. About 40 mg samples were firstly loaded into the reactor and pretreated at 300 °C for 0.5 h in 99.99% Ar flow (30 ml min$^{-1}$). After cooling down to room temperature, the samples were heated in a 5% H$_2$/Ar (30 ml min$^{-1}$) flow to 500 °C with a heating rate of 10 °C min$^{-1}$. The amount of exposed Cu was measured by N$_2$O oxidation followed by the TPR method[50] which was carried out in a homemade setup[51]. After reduction treatment at 500 °C for 1 h in pure H$_2$ (30 ml min$^{-1}$), the sample was swept in a 5% N$_2$O/He (30 ml min$^{-1}$) atmosphere at 60 °C for 1 h to oxidize exposed surface Cu atoms into Cu$_2$O. Then TPR was performed as before to measure the amount of Cu$_2$O, and the consumed H$_2$ was calibrated by pure CuO (99.99%) with a specific amount. Copper dispersion was calculated by dividing the number of exposed copper atoms by the total number of supported copper atoms per gram of the catalyst. Exposed copper surface area was calculated based on an atomic copper surface density of 1.46 × 10$^{19}$ Cu atoms m$^{-2}$. The amount of Cu loading was identified by inductively coupled plasma-optical emission spectrometry (ICP-OES). Powder X-ray diffraction (XRD) patterns were recorded on an X'Pert Pro (PANalytical) diffractometer with Cu Ka radiation at 40 kV and 40 mA.

**XAS**. X-ray absorption (XAS) data up to $k = 16$ Å$^{-1}$ at Cu K (8979 eV) absorption edge were recorded at room temperature in transmission mode with a step size of 0.5 eV on catalysts and reference Cu metal foil at beamline BL14W1 of the Shanghai Synchrotron Radiation Facility (SSRF) in China[52]. Each Cu K-edge extended X-ray absorption fine structure (EXAFS) spectra were recorded for 7 min. CuO and Cu$_2$O were pressed into pellets. All the tested samples (about 20 mg) were ground into a fine powder and uniformly coated on tape in the air. The time-resolved XAS spectrum was analyzed by linear combination analysis (LCA) with three standard spectra: Cu foil, Cu$_2$O, and CuO. EXAFS data were processed and analyzed using the IFFEFIT software package[53]. Artemis was used to fit the experimental data with model structures of metallic Cu (ICSD 43493) and CuO (ICSD 16025) from the Inorganic Crystal Structure Database (ICSD). Structural parameters include energy shift of the path ($\Delta E_0$), change in the half path length ($\Delta R$), amplitude reduction factor ($S_0^2$), coordination number (CN), and relative mean-square displacement of the atoms included in the path (Debye–Waller factor, $\sigma^2$).

**DFT methods**. DFT calculations were performed by using the Vienna ab-initio simulation package (VASP). The generalized gradient approximation in the Perdew–Burke–Ernzerhof form was used to describe the exchange and correlation energy. The model of Cu was built with Cu$_{25}$ cluster loading on TiO$_2$ layers. Cu$_{25}$ model was chosen because its dimension of 2–3 nm in length, width and height are very close to our Cu nanoparticles. Three catalyst models were established for comparison, including Cu$_{25}$ clusters loaded on TiO$_2$ support without encapsulation (denoted as Cu$_{25}$/TiO$_2$), as well as a TiO$_x$ layer partially coated on Cu$_{25}$ clusters on both TiO$_2$ and La-doped TiO$_2$ supports (denoted as Cu$_{25}$@TiO$_x$/TiO$_2$ and Cu$_{25}$@TiO$_x$/LaTiO$_2$, respectively). As for the model of Cu$_{25}$@TiO$_x$/TiO$_2$ and Cu$_{25}$@TiO$_x$/LaTiO$_2$, the loaded Cu$_{25}$ clusters were part-coated by one TiO$_2$ layer. During structure optimization, the bottom TiO$_2$ layers were fixed, and the Cu cluster with TiO$_2$ coat was relaxed. The cutoff energy of the plane-wave basis set was 400 eV. As for geometric optimization, the atomic positions were optimized until the forces were less than 0.05 eV Å$^{-1}$ and 3 × 3 × 1 Monkhorst-Pack sampled k-points were used in all slabs systems. To avoid interactions between neighbor slabs caused by the periodic boundary conditions, the vacuum spacing perpendicular to the surface was 15.0 Å. The adsorption energy is described as follows: $\Delta E = E_{ads} - E_{slab} - E_{molecular}$. $\Delta E$ is adsorption energy. $E_{ads}$ is the calculated energy

of slab with intermediates. $E_{slab}$ and $E_{molecular}$ are the energy of slab and molecule, respectively. The transitional state (TS) was located using the climbing image Nudged Elastic Band method.

## Data availability

The main data supporting the findings of this study are available within the paper and its Supplementary Information. Additional data are available from the corresponding authors upon reasonable request.

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

## Acknowledgements

We would like to acknowledge Dr. Peipei Zhang at the University of Toyama for her help in sputtering experiments, Dr. Jinsuo Gao at the Dalian University of Technology for his help in EPR measurements, Prof. Xi Liu at Shanghai Jiao Tong University for his fruitful discussion in TEM and EELS analyses, Mr. Hiroaki Matsumoto at Hitach High-Tech for his assistance in STEM experiments at Hitachi HF5000 microscope and Dr. Xuefeng Chu for his kindly help with XPS analysis. We also appreciate beamline BL14W1 in Shanghai Synchrotron Radiation Facility (SSRF) for providing the beam time. This study was supported by the Liaoning Revitalization Talents Program (XLYC1907066 and XLYC1907053), Dalian Outstanding Young Scientific and Technological Talents Program (2018RJ06), the Youth Innovation Promotion Association of Chinese Academy of Sciences (2018214 and 2018220), and the National Natural Science Foundation of China

(22078315, 22172169, and 21872144). Financial aids from JST of Japan and YKK company are greatly appreciated.

## Author contributions

J.Y. and J.S. conceived and designed this work. J.Y. conducted the characterization of catalysts, analyzed the data, and wrote the manuscript. X.S., X.T., and J.Z. prepared the catalysts and tested the catalytic performance. J.L. contributed to the SP preparation. S.L. and Y.L. performed the TEM characterizations and analysis. N.T. and T.A. supplied the SP technique. J.S. was in charge of the project and contributed to the interpretation of data. All corresponding authors discussed the manuscript.

## Competing interests

The authors declare no competing interests.
