## [Peer Review File · Nature Communications]

Title: Ultra-high Thermal Stability of Sputtering Reconstructed Cu-based CatalystsREVIEWER COMMENTS

Reviewer #1 (Remarks to the Author):

The submitted manuscript - Ultra-high Thermal Stability of Sputtering Reconstructed Cu-based Catalysts provides experimental studies about highly stable Cu-based catalyst in order to serve a general guide to designed stable heterogeneous catalyst even at higher temperatures. The authors compare synthesised Cu particles encapsulated either in LaTiO₂ or commercial TiO₂ as a support of the catalyst. However from their results can be seen that only in case of LaTiO₂ support Cu particles was encapsulated. The catalysts were tested on the reverse water gas shift reaction (RWGS) at temperature 600°C where showed extremely stable performance for 500h and in case of Cu/LaTiO₂ stable even at 800°C for 100h. Moreover the manuscript combined experimental and theoretical research due to the fact that it uses the DFT method in order to study molecule-level mechanism of RWGS reaction.

From the grammatical part the level of language used is very good. However in the part Main, first paragraph, sentence “As a result, the operating temperatures of must be restricted...” – there is missing word- “of RWGS” I assume. From the bibliography part authors used only current literature with the required amount of literature. From the formal part I suggest to make a table in which will be summary comparison of both support include the catalytic performance, SMSI and surface differences in order to make the article clearer.

The submitted manuscript provides interesting high-quality results which are supported by both experimental research and theoretical study and I can recommend for publication only. Very good article.

Reviewer #2 (Remarks to the Author):

In the manuscript, the authors claim promoting the SMSI effect for Cu/LaTiO₂ nanoparticles, which is able to improve the thermal stability for the Cu nanoparticles due to the oxide layer formed over the Cu nanoparticles during reduction treatment. The authors claim tailoring the extent of the SMSI effect by the method of synthesis of the catalysts. I'm not convinced about most of the results shown, where there are misinterpretation in several cases. Some specific points:

- 1) What about the size of the Cu nanoparticles before thermal treatment? There is no information about that, then it is not possible to affirm the absence of sintering due to thermal treatment.
- 2) Figure S1(d) does not show clearly the Cu nanoparticles for making the histogram of size distribution. There is also a mistake because in the image the mean size is 2.3 nm but in the text it is 2.8 nm. Also, there is no indication about the number of Cu nanoparticles used in the histogram. What is the standard deviation of the histograms? What was the criterion used for determining the Cu nanoparticles size?
- 3) The authors comment about the absence of the classical SMSI effect in Cu-based nanoparticles before sintering occurrence. The authors also affirm the existence of some few large particles after thermal treatment. However, the encapsulation layer is observed only for the bigger particles, like that shown in Figure 1(c) and 1(e), Figure S2, and Figure S10. Even the smaller particle with the oxide layer (Figure 1(c))

is 3-4 times bigger than the original particle. Then it is not clear from TEM analysis that the authors were able to promote the oxide layer over the small Cu nanoparticles. How do the authors are confident on the existence of the oxide layer over the majority of the nanoparticles, which are small particles, where the authors did not identify the oxide layer? The fact they “came up” in TEM images after oxidation treatment is not a convincing proof.

4) Regarding the FT of the EXAFS oscillations at Figure 1i, the best way to identify the chemical compound from XAS analysis is using the XANES data. There will be possible to quantify about the amount of Cu(0), Cu₂O and CuO in the samples. Formally, the FT analysis does not prove the existence of a CuO compound, it only shows the presence of some light element at this distance. Furthermore, why SP-Cu/LaTiO₂-500R sample does not present a Cu-O scattering if the oxide layer is covering the Cu nanoparticle and, then, it should be a Cu-O scattering path there?

5) What was the time spent for each sample between air exposition after thermal treatment and EXAFS measurements shown in Figure 1i?

6) The authors claim identifying the Ti-O-La chemical component in the Ti 2p XPS spectrum (Figure S5). However, it is not clear why this component is located in a higher binding energy than the TiO₂ component neither a reference is used to justify such identification.

7) The La 3d XPS spectrum at Figure S5 shows 3 components, while in the text it is discussed the existence of only 2 components. What about the component at 834.9 eV?

8) What is the amount of rutile and anatase phases? Where is La located? Why?

9) The sample with the SMSI effect is active for the RWGS reaction. Why does it occurs if the catalytic active sites at Cu surface are covered by the oxide layer?

10) The oxide layer is formed after reduction treatment. What happens during RWGS (mixed CO₂ + H₂ atmosphere) regarding the oxide layer? Is it still there?

11) Another point is about the observation of reduction in the mean size from 2.8 nm to 0.9 nm in the first 100 h and then to 1.5 nm at the end of 500 h (Figure 2(b)-(d)). The same for Figure 2(f). This reduction from 2.8 nm to 0.9 nm is typical of sintering by Ostwald ripening process, which can explain the existence of bigger nanoparticles of around 10 nm size. The authors claim it occurs because the nanoparticles are redispersed by SMSI like that in reference 21 but there the dispersion is induced by carbonization. Then it seems the occurrence of sintering by Ostwald ripening here instead of the thermal stability claimed by the authors. Furthermore, the image quality is poor and opens doubts about the affirmation that there are no changes on shape and size.

12) Figure S3 is also not convincing about the presence of the oxide layer surrounding the Cu nanoparticles. Firstly, whether the Cu nanoparticle is covered by TiO_x, as stated in the paper, I would expect to see a stronger Ti signal. Furthermore, what is the noise level of the measurement? The Ti signal shows 2 counts at the region of interest, which is probably the same value for the noise level. The authors should include the EDS line scan further away from the nanoparticle in order to determine the noise level.

13) It is well known that the electron transfer between support and nanoparticle occurs before the migration of oxides from support to the nanoparticle surface (S. Bernal et al, Catal. Today. 1999, 50, 175), that is, at smaller temperatures. Then it is not expected to observe the energy shifts in the in-situ XPS spectra shown in Figure 3b. At least, it is not expected to observe both oxide layer and charge transfer for the same thermal treatment in the classical SMSI effect.

14) Why do the authors observe Cu 2p XPS spectra with almost the same intensity at Figure 3 if the sample after reduction treatment is covered by an oxide layer? The Cu 2p XPS intensity should decrease significantly in this case.

15) It is not clear the existence of Cu(0) instead of Cu₂O from both XPS/AES and XAS data. The Cu L₃M_{4,5}M_{4,5} Auger spectrum from Cu(0) presents 4 distinct contributions and that from Cu₂O presents 2 main contributions. See the corresponding figures of the cited classical references (Surf. Sci. Spectra, 1993, 2, 55 for Cu₀ and Surf. Sci. Spectra, 1998, 5, 257 for Cu₂O). The Auger spectrum enables easy distinction of Cu₀ to Cu₂O by the shape of the Auger spectrum, which is consistent with the presence of Cu₂O in this work (see Figure S17). Regarding XAS data, the XANES analysis are not shown by the authors for these samples. It is only shown a comparison with the Cu(0) and CuO standards but it is not shown the XANES spectrum for the Cu₂O standard. From the fingerprint existing at XANES of Cu₂O in the literature, I would expect a combination of both Cu(0) and Cu₂O compounds from the XANES analysis. The procedure adopted for fitting the FT is not shown and the fit quality is shown only for 2 cases.

16) Figure 3 and S16 are not convincing in the XPS part. The Ti 2p XPS data are shown without the fit of the data. The Cu 2p XPS spectra are not properly fitted since there is a component at around 931.7 eV in the ROR case that is not included in the fit and the components used are not properly identified because there are missing identification. The fit procedure should be described and the fit result should be included. Furthermore, why the XPS spectrum of SP-Cu/LaTiO₂-ROR sample is different than that of SP-Cu/LaTiO₂-500R samples but such difference is not observed in the AES spectra?

17) The TiO₂ support may present sample charging effects during XPS measurements. How it was handled by the authors?

18) The authors should include the Survey XPS spectra showing the absence of contamination.

19) Table S2: Cu loading is not the same between the samples, how do the authors compare the catalytic results?

20) It was not measured in-situ XPS spectra because, after treating the sample inside the XPS chamber, the XPS spectra were "measured when cooling down to room temperature". In-situ XPS spectra would be measured at the thermal treatment temperature and with the sample exposed to the gas. By the way, were the XPS spectra measured in vacuum during cooling? What was the pressure?

21) No details are given about the sample preparation for the measurements performed. Also, no details are given about the procedure adopted to analyze the data obtained, specially XPS and XAS spectra.

Reviewer #3 (Remarks to the Author):

It is a very interesting paper proposing a novel method for preparing high stability Cu-based catalysts with SMSI. The paper is easy reading and very well organised. Many modern techniques have been used for the characterisation of the samples while the main conclusions are well supported by the experimental results. In my opinion the manuscript can be published after minor revisions on the following points which need clarification/discussion.

Introduction: Authors should discuss in more detail two aspects, the mechanism of the reconstruction of the metal electronic structure using the sputtering technique and how the reducibility of the support enhances the capability of the metal for hydrogen dissociation. A few lines are written in the Results however discussion is also necessary in Introduction.

Results: Please explain why La enhances the reducibility of the TiO₂.

Results-Figure S2. Is the term dynamic capsulation process the appropriate one ? Is this an in-situ experiment ? The examined area of the sample (in these 3 images) is always the same ?

Results-Figure 2a. It is really amazing the stability of the SP-Cu/LaTiO₂ catalyst. It is also very interesting the initial increase of the rate (during the first 100 h). According to the authors this is due to the decrease of the size of the particles. Please give a possible mechanism explaining how a possible metal re-dispersion at 600 °C can lead in metal particle decrease.

Results-page 13 (lines 1-4). Authors give DFT calculations based on a specific mechanism (RWGS reaction pathway). However, other possible steps may be also possible (like CO₂ dissociative adsorption).

Do the authors exclude any catalytic contribution of La ?

Point-by-point responses to the comments

Manuscript ID: NCOMMS-21-26046-T

Title: Ultra-high Thermal Stability of Sputtering Reconstructed Cu-based Catalysts

Dear reviewers,

We are very grateful to you for the very valuable and insightful comments on our manuscript. It is a very good opportunity to help us improve the whole quality of the manuscript. Herein, all of these comments are carefully answered and additional experiment data are provided. In the revised version, the unique SP-Cu/LaTiO₂ series catalysts were systematically investigated *via* supplying new results and more discussions in depth, such as HRTEM, XPS, XANES etc. We believe that the revised manuscript provides a clearer and in-depth insight on ultra-high thermal stability of sputtering reconstructed Cu-based catalysts, which will be meaningful to wide catalysis and material researchers. The following descriptions are the responses to these excellent comments one by one.

- ✓ Response to reviewer 1: Page 2-3;
- ✓ Response to reviewer 2: Page 4-24;
- ✓ Response to reviewer 3: Page 25-28.

Thanks again for your reviewing.

Kind regards,

Prof. Dr. Jian Sun

on behalf of co-corresponding authors, Prof. Noritatsu Tsubaki & Prof. Yuefeng

Liu

Reviewer #1 (Remarks to the Author):

The submitted manuscript - Ultra-high Thermal Stability of Sputtering Reconstructed Cu-based Catalysts provides experimental studies about highly stable Cu-based catalyst in order to serve a general guide to designed stable heterogeneous catalyst even at higher temperatures. The authors compare synthesized Cu particles encapsulated either in LaTiO₂ or commercial TiO₂ as a support of the catalyst. However from their results can be seen that only in case of LaTiO₂ support Cu particles were encapsulated. The catalysts were tested on the reverse water gas shift reaction (RWGS) at temperature 600°C where showed extremely stable performance for 500h and in case of Cu/LaTiO₂ stable even at 800°C for 100h. Moreover the manuscript combined experimental and theoretical research due to the fact that it uses the DFT method in order to study molecule-level mechanism of RWGS reaction.

From the grammatical part the level of language used is very good. However in the part Main, first paragraph, sentence “As a result, the operating temperatures of must be restricted...” – there is missing word- “of RWGS” I assume.

Response: We appreciate very much for the comment. It was a type mistake. We revised the sentence into “As a result, the operating temperatures for those catalysts must be restricted...”, because the operating temperature should be limited below 300°C for all Cu catalyzed reaction, including RWGS.

From the bibliography part authors used only current literature with the required amount of literature. From the formal part I suggest to make a table in which will be summary comparison of both support include the catalytic performance, SMSI and surface differences in order to make the article clearer.

Response: Thank you for the constructive suggestion. The basic information, such as Cu loading, Cu surface area, Cu dispersion and reaction rate were summarized in Table

S3 in the revised supplementary materials to make the article clearer.

Table S3 | Comparison of Cu property and activity on various TiO₂ support.

Property of Cu in IM-Cu/TiO₂, SP-Cu/TiO₂ and SP-Cu/LaTiO₂ after being pretreated in H₂ at 500 °C for 1 h (-500R).

Catalysts	Cu loading ^a (wt.%)	Exposed Cu surface area ^b (m ² /g)	Cu dispersion (%)	Reaction rate ^c (mmol _{CO₂} /m ² Cu·h)	TOF ^d (s ⁻¹)
IM-Cu/TiO ₂ -500R	12.6	28.8	35.4	1.9	0.02
SP-Cu/TiO ₂ -500R	18.7	28.3	23.5	9.3	0.11
SP-Cu/LaTiO ₂ -500R	14.6	12.7	13.5	1.9	0.05

^aCu loading was identified by XPS.

^bThe exposed Cu surface area was measured by TPR after N₂O oxidation at 60 °C for 1 h.

^cThe reaction rate was calculated according to the CO₂ conversion at 2 h. The metallic Cu surface area used here was computed based on an atomic copper surface density of 1.46×10^{19} Cu atoms per m².

^dTOF was calculated based on the reaction rate measured under kinetics conditions at 1h.

The submitted manuscript provides interesting high-quality results which are supported by both experimental research and theoretical study and I can recommend for publication only. Very good article.

Response: We appreciate very much for the comment.

Reviewer #2 (Remarks to the Author):

In the manuscript, the authors claim promoting the SMSI effect for Cu/LaTiO₂ nanoparticles, which is able to improve the thermal stability for the Cu nanoparticles due to the oxide layer formed over the Cu nanoparticles during reduction treatment. The authors claim tailoring the extent of the SMSI effect by the method of synthesis of the catalysts. I'm not convinced about most of the results shown, where there are misinterpretation in several cases. Some specific points:

1) What about the size of the Cu nanoparticles before thermal treatment? There is no information about that, then it is not possible to affirm the absence of sintering due to thermal treatment.

Response: Thanks for the valuable comments. We have provided the morphology of Cu nanoparticles before thermal treatment (Fig. R1) with Aberration-corrected HRTEM technique. As far as we understand, Cu nanoparticles before thermal treatment are those in fresh sample. Usually, metal particles were identified by different contrast from other components or lattice fringes of specific crystal plane. However, the crystallinity of metallic Cu was low, because sputtered Cu atoms were deposited on the surface of support by gravity and gathered to small particles without calcination process in SP method, leading to poor contrast of Cu nanoparticles in a TEM view and absence of lattice fringes. As can be seen in the fresh sample (below), a few humps with a particle size of around 1 nm existed on the surface of support particles. But they were hard to be clearly identified by either lattice fringes or image contrast. Although we cannot affirm the absence of sintering during the reduction process before the formation of SMSI and encapsulation, this process will not affect the stability of Cu catalyst after SMSI formation. The discussion was added in the revised manuscript on Page 6.

Fig. R1 HRTEM images of the fresh sample (a) SP-Cu/LaTiO₂-fresh and (b) magnification of the marked area.

2) Figure S1(d) does not show clearly the Cu nanoparticles for making the histogram of size distribution. There is also a mistake because in the image the mean size is 2.3 nm but in the text it is 2.8 nm. Also, there is no indication about the number of Cu nanoparticles used in the histogram. What is the standard deviation of the histograms? What was the criterion used for determining the Cu nanoparticles size?

Response: We appreciate very much for the comment. The mistake of 2.8 nm in the main text was revised into 2.3 nm. The average Cu particle size and related standard deviation of the histogram were counted from more than 100 particles of each sample. Although the crystallinity of metallic Cu was improved after reduction at 500°C, Cu nanoparticles in SP-Cu/LaTiO₂-500R sample were still hard to be identified by either lattice fringes or image contrast (the reason will be discussed in the next response). So we applied the ADF model (STEM images) instead of transition model (TEM images) with high acquisition time for higher spatial resolution of Cu nanoparticles (new Fig. S3) [Zhang et al., *ChemCatChem* 2011, 3, 965 - 968]. The intensity difference between Cu and the support can be clearly observed, giving better identification of the boundary

of particles. The diameters of each sphere particles were measured. The statistical size distribution of those particles was shown in the form of particle numbers with size falling in a specific range.

3) The authors comment about the absence of the classical SMSI effect in Cu-based nanoparticles before sintering occurrence. The authors also affirm the existence of some few large particles after thermal treatment. However, the encapsulation layer is observed only for the bigger particles, like that shown in Figure 1(c) and 1(e), Figure S2, and Figure S10. Even the smaller particle with the oxide layer (Figure 1(c)) is 3-4 times bigger than the original particle. Then it is not clear from TEM analysis that the authors were able to promote the oxide layer over the small Cu nanoparticles. How do the authors are confident on the existence of the oxide layer over the majority of the nanoparticles, which are small particles, where the authors did not identify the oxide layer? The fact they “came up” in TEM images after oxidation treatment is not a convincing proof.

Response: This is a very good comment.

Firstly, the existence of the oxide layer over the majority of the small Cu nanoparticles can be deduced from the results of electron microscopy. Cu particles of SP-Cu/LaTiO₂-500R sample were unable to be observed on TEM images (Fig. S2) but can be clearly seen on STEM images (new Fig. S3 in the revised supplementary materials), confirming the existence of Cu particles. Usually, metal particles were identified by different image contrast from other components or lattice fringes of specific crystal plane. The poor contrast caused by similar atomic-number of Cu and Ti ($Z_{\text{Cu}} = 29$ vs. $Z_{\text{Ti}} = 22$) can be excluded, because Cu particles can be clearly observed on SP-Cu/TiO₂-500R in Fig. S1b. The preparation method and the pretreatment process

of SP-Cu/LaTiO₂-500R and SP-Cu/TiO₂-500R were completely the same, except the doping of 5% La atoms. The effect of La with much larger atomic-number ($Z_{\text{La}} = 57$) in the support on the poor contrast can also be excluded since Cu particles appeared after oxidation treatment as seen in Fig. S1c for SP-Cu/LaTiO₂-ROR sample. This phenomenon revealed that the distinguish of Cu and LaTiO₂ support was practicable. Therefore, in this system, the indistinguishable species were only Ti oxides layer and LaTiO₂ support. The reduction of Cu contrast was caused by the coverage of Ti oxides layer on the surface of Cu particles, which made them indistinguishable. From the analysis above, the particles can only be distinguished by lattice fringes. Large Cu particles were easily identified by lattice fringes even with oxides layer on the surface as shown in Fig. 1e,f. So the large Cu particles were used to show the coverage layers, because those oxide layers on the surface of major small Cu particles around 2-3 nm were unable to be clearly observed by TEM even with one of the most advanced techniques (the special aberration corrector equipped below the objective lens with TEM resolution better than 0.1 nm).

Secondly, the encapsulation on major Cu nanoparticles can be further confirmed by CO adsorption. The blocked access of small molecules after reduction was one of the main characters of classical strong metal-support interaction, which caused by the formation of coverage layer on the surface of metallic Cu. In Fig. 3a, the CO-IR signal was clearly decreased after reduction at 500°C. Compared to the local images from electron microscopy, CO-IR can give more macroscopic information, indicating the existence of the oxide layer over the majority of the small Cu nanoparticles.

The new Fig. S2 was added in the revised supplementary materials. The discussion was added on Page 6 of the revised manuscript.

Figure S1 | Distribution of Cu nanoparticles in different samples. TEM images and particles size distributions of (a) IM-Cu/TiO₂-500R, (b) SP-Cu/TiO₂-500R and (c) SP-Cu/LaTiO₂-ROR. The average particle size of Cu in different samples are 2.3, 3.0, 2.8 nm, respectively.

Figure S2 | Identification of Cu nanoparticles. HRTEM images of SP-Cu/LaTiO₂-500R.

4) Regarding the FT of the EXAFS oscillations at Figure 1i, the best way to identify the chemical compound from XAS analysis is using the XANES data. There will be possible to quantify about the amount of Cu(0), Cu₂O and CuO in the samples.
Formally, the FT analysis does not prove the existence of a CuO compound, it only

shows the presence of some light element at this distance. Furthermore, why SP-Cu/LaTiO₂-500R sample does not present a Cu-O scattering if the oxide layer is covering the Cu nanoparticle and, then, it should be a Cu-O scattering path there?

Response: Thanks for the valuable comments. We completely agreed with the reviewer about the inappropriate analysis of CuO compound by FT of EXAFS spectra. The old Fig. 1i of Fourier transforms of k^3 -weighted Cu K EXAFS signals was replaced by XANES data, where the component of Cu species was further analyzed by linear combination fitting. The results were added in supplementary materials as Table S1. Although about 30% CuO and Cu₂O species were found in SP-Cu/LaTiO₂-500R sample, most of Cu species were still stayed at metallic state. It indicated that most of Cu particles were covered by oxide layer, considering that no metallic Cu can exist in the air without protective layer as the other three samples. The discussion was added on Page 8 in the revised manuscript).

For the second question, Cu-O scattering was observed on SP-Cu/LaTiO₂-500R sample as shown in Table S2 in the revised supplementary materials.

New Fig. 1i Cu K-edge X-ray absorption near edge structure (XANES) of different samples, Cu foil and reference CuO.

Table S1 | Analysis of Cu component. Linear combination fitting results of Cu K XANES data of IM-Cu/TiO₂-500R, SP-Cu/TiO₂-500R, SP-Cu/LaTiO₂-500R and SP-Cu/LaTiO₂-ROR samples.

	Cu (%)	Cu ₂ O (%)	CuO (%)	R factor
IM-Cu/TiO ₂ -500R	0.0±3.8	11.3±4.2	88.5±2.1	0.006
SP-Cu/TiO ₂ -500R	0.0±3.7	6.9±4.1	92.6±2.1	0.006
SP-Cu/LaTiO ₂ -500R	70.1±2.0	17.2±2.2	14.4±1.1	0.002
SP-Cu/LaTiO ₂ -ROR	0.0±3.5	8.7±3.8	90.8±1.9	0.005

5) What was the time spent for each sample between air exposition after thermal treatment and EXAFS measurements shown in Figure 1i?

Response: The exposition time are not exactly the same for different samples, but they are under similar time range (several weeks). Therefore, the effect of exposition time can be well excluded.

6) The authors claim identifying the Ti-O-La chemical component in the Ti 2p XPS spectrum (Figure S5). However, it is not clear why this component is located in a higher binding energy than the TiO₂ component neither a reference is used to justify such identification.

Response: Thank you for this very helpful suggestion.

For the identification of Ti-O-La species, the main proof was the peak at 834 eV in La 3d spectra that has been reported in Ref. 31 (*Appl. Catal. B: Environ.* 2015, 168-169, 125-131.). It was also reported that peak at 532.2 eV in O 1s spectra can be assigned to Ti-O-La according to Ref. 34 and 35 (*The Journal of Physical Chemistry B* 2002, 106,

5695-5700; *Electrochim. Acta* 2015, 153, 170-174.). We inferred that the extra peak at 459.8 eV was derived from the Ti-O-La species compared to standard TiO₂ reference, because the binding energy of Ti 2p_{3/2} shifted to higher energy when adding La³⁺ as reported (*Appl. Surf. Sci.* 2008, 254, 7314-7320, which was added as Ref. 36 in the revised manuscript).

The discussion on La and O was moved before Ti in the revised manuscript for clear understanding. And the discussion was revised as “*Typical peaks of Ti 2p_{3/2} and Ti 2p_{1/2} could be detected at 458.3 and 464.2 eV, respectively. The splitting between the two core levels was close to TiO₂ reference (5.7 eV), indicating a valence state of +4 for Ti in the support. Compared to SP-Cu/TiO₂ sample, an extra peak at 459.8 eV was found, which was probably related to Ti–O–La species, because it has been reported that the binding energy of Ti 2p_{3/2} shifted to higher energy when adding La³⁺ species³⁶.*”

7) The La 3d XPS spectrum at Figure S5 shows 3 components, while in the text it is discussed the existence of only 2 components. What about the component at 834.9 eV?

Response: We appreciated this valuable question very much.

According to XPS results in Supplementary Fig. S7, a splitting of the La 3d_{5/2} and La 3d_{3/2} lines was observed in the La 3d core level spectra. For La 3d_{3/2}, the peaks at 851.7 and 855.4 eV were identified as the main and the shake-up satellite peaks, respectively. The binding energy difference between the main and satellite peaks (ΔE) was 3.7 eV. Accordingly, ΔE should be the same for La 3d_{5/2}, i.e., the main and the shake-up satellite peaks should be at 834.8 and 838.5 eV, respectively. In addition to the peak position and splitting, ΔE in the multiples split can be used to distinguish La₂O₃ from other La³⁺

compounds. Usually, the ΔE values for La^{3+} compounds are in the range of 3.5 to 4.6 eV for the La 3d_{5/2} spectrum, where the one for La_2O_3 was 4.6 eV. Therefore, this La XPS profile revealed the presence of a mixture of lanthanum oxide species. Furthermore, we found that the one at 834.8 eV disappeared and split into two peaks, which can be assigned to Ti–O–La species (834.0 eV³¹) and La_2O_3 (835.4 eV³²), indicating that Ti–O–La species were co-existing with pure La_2O_3 in SP-Cu/LaTiO₂ sample. The Fig. S7 in the revised supplementary materials was changed and the detailed discussion was added on Page 10 according to the above two responses.

Figure S7 | Analysis of Ti-O-La structure. Survey (a), La 3d (b), O 1s (c) and Ti 2p (d) XPS spectra of SP-Cu/LaTiO₂-fresh sample.

8) What is the amount of rutile and anatase phases? Where is La located? Why?

Response: For the first question, we use the commercial P25 (Degussa) type TiO₂

powder as the support of IM-Cu/TiO₂ and SP-Cu/TiO₂ samples. The amount of rutile and anatase phases were calculated by eq. $f_A=1/(1+1.26I_{R(110)}/I_{A(101)})$, where f_A is the amount of anatase in TiO₂; $I_{A(101)}$ and $I_{R(110)}$ are the peak intensity of anatase (101) and the rutile (110) in XRD, respectively. The ratio of anatase to rutile calculated by the XRD pattern in Fig. S8 was 82%, which was well consistent with the information from the reagent (80%). LaTiO₂ support was made by FSP method, where the ratio of anatase to rutile was 62%, which might be caused by the high temperature process in FSP method. This discussion was added on Page 10 in the revised manuscript and the supplementary materials.

For the second question, from the ADF-STEM images in Supplementary Fig. S9, the bright particles in SP-Cu/LaTiO₂ sample with a size about 2 nm marked by red circles can be assigned to Cu nanoparticles, while the bright dots marked by yellow circles can be identified as La atoms compared to SP-Cu/TiO₂, indicating that La was atomically dispersed in TiO₂ lattice. The discussion of La location was added on Page 11 in the revised manuscript.

Figure S8 | Analysis of phase component. XRD patterns of the TiO₂ supports: TiO₂ nanoparticles (TiO₂) and 3.8wt.% La doped TiO₂ made by FSP (LaTiO₂). Usually, the amount of rutile and anatase phases were calculated by eq. $f_A=1/(1+1.26I_{R(110)}/I_{A(101)})$, where f_A is the amount of anatase in TiO₂; $I_{A(101)}$ and $I_{R(110)}$ are the peak intensity of anatase (101) and the rutile (110) in XRD, respectively. The ratio of anatase and rutile was 82% and 62 % for TiO₂ and LaTiO₂ support, respectively.

9) The sample with the SMSI effect is active for the RWGS reaction. Why does it occurs if the catalytic active sites at Cu surface are covered by the oxide layer?

Response: This is a very good and interesting question. We have successfully designed a lot of work to figure out what happened on the surface during reaction. Actually, the coverage layer was partially removed by H₂O oxidation during reaction, exposing part of the catalytic active sites at Cu surface to reach a well dynamic balance for the catalysis of reaction. More details were also available in the response of the next question.

10) The oxide layer is formed after reduction treatment. What happens during RWGS (mixed CO₂ + H₂ atmosphere) regarding the oxide layer? Is it still there?

Response: It has been reported that the overlayer derived from the SMSI between Rh and TiO₂ can be oxidized in the humid environment of the CO₂+H₂ reaction³⁹. In our case, the extent of encapsulation and the exposure of Cu active sites were also found in a dynamic equilibrium process during reaction in the combined effects of H₂ reduction and H₂O oxidation. Thus, NO reduction by CO to produce N₂ and CO₂ was performed as a probe reaction after H₂O and H₂ treatments to evaluate the variation of coverage as shown in Supplementary Fig. S13. After pretreatment in H₂ (600H), Cu nanoparticles

were fully covered by TiO_x species, leading to complete deactivation. Then, the sample was pretreated in 15% $\text{H}_2\text{O}/\text{He}$ atmosphere at 600°C for 1h. CO conversion increased to 71.37% but was still lower than the initial conversion (89.13%), indicating that the coverage was partially removed by H_2O oxidation. Afterwards, the sample was further pretreated in 15% $\text{H}_2\text{O}/\text{H}_2$ atmosphere at 600°C for 1h. CO conversion decreased to 39.04%, demonstrating that the removed coverage can be partially recovered by addition of H_2 in H_2O atmosphere. The combined reduction and oxidation effects in the system went through a dynamic equilibrium process until the coverage extent got a stable state. Accordingly, the exposed Cu surface area decreased to $12.7 \text{ m}^2\cdot\text{g}^{-1}$ after reduction and recovered to $35.4 \text{ m}^2\cdot\text{g}^{-1}$ after reaction. When the reaction stopped, the entire encapsulation formed again in H_2 protection atmosphere as observed in Supplementary Fig. S14-16. This part of discussion was put in Page 12 of the revised manuscript.

11) Another point is about the observation of reduction in the mean size from 2.8 nm to 0.9 nm in the first 100 h and then to 1.5 nm at the end of 500 h (Figure 2(b)-(d)). The same for Figure 2(f). This reduction from 2.8 nm to 0.9 nm is typical of sintering by Ostwald ripening process, which can explain the existence of bigger nanoparticles of around 10 nm size. The authors claim it occurs because the nanoparticles are redispersed by SMSI like that in reference 21 but there the dispersion is induced by carbonization. Then it seems the occurrence of sintering by Ostwald ripening here instead of the thermal stability claimed by the authors. Furthermore, the image quality is poor and opens doubts about the affirmation that there are no changes on shape and size.

Response: Thank you for this good comment. Generally, the Ostwald ripening is a process by which components of the discontinuous phase diffuse from smaller to larger droplets through the continuous phase. While the re-dispersion phenomenon of metal nanoparticles is one of the classical features of SMSI, which is widespread on various

metals, such as Ir/TiO₂ (*Applied Catalysis B: Environmental*, 2021, 297: 120410) and Rh/TiO₂ (*Journal of Environmental Chemical Engineering*, 2021, 9: 105790), because large amount of oxygen vacancies was produced on the reducible support during high temperature reduction process. Meanwhile, the migration of Cu atoms from the Cu nanoparticles to the support will happen during the thermal motion process. These Cu atoms will be captured and settled by the oxygen vacancies under the SMSI, inducing the re-dispersion of metal particles. In our case, the Cu particles were firstly observed to get smaller in the initial 100h of time on stream, then gradually grow up and finally reach a stable state during 500h reaction. According to the evolution of particles size, the Ostwald ripening process and re-dispersion phenomenon probably co-exist in the whole reaction process in a restrictive relation. The re-dispersion function was stronger than Ostwald ripening in the first 100h, then the particles grew up via Ostwald ripening process until they reached a balance and kept the particle size stable.

More references and discussions about the re-dispersion function of SMSI were added on Page 12 of the revised manuscript.

12) Figure S3 is also not convincing about the presence of the oxide layer surrounding the Cu nanoparticles. Firstly, whether the Cu nanoparticle is covered by TiO_x, as stated in the paper, I would expect to see a stronger Ti signal. Furthermore, what is the noise level of the measurement? The Ti signal shows 2 counts at the region of interest, which is probably the same value for the noise level. The authors should include the EDS line scan further away from the nanoparticle in order to determine the noise level.

Response: This is a very helpful question. As can be seen in the revised Supplementary Fig. S5, the EDS line was prolonged to 1.4 nm. The highest noise level of Ti signal was

about 1 counts. Many points of Ti signals above 2 counts can be observed at the position of the surface of Cu particles, confirming the existence of Ti species coverage on at least half of the Cu surface.

Figure S5 | Identification of encapsulation. ADF-STEM image (a) and line scanning elemental analysis (b) of SP-Cu/LaTiO₂-500R sample. The line profile was collected along the yellow arrow in the image. Ti species could be detected on the Cu surface, indicating that at least half of the 0.8 nm Cu nanoparticle was encapsulated by TiO_x species as the profile described in the inset image.

13) It is well known that the electron transfer between support and nanoparticle occurs before the migration of oxides from support to the nanoparticle surface (S. Bernal et al, Catal. Today. 1999, 50, 175), that is, at smaller temperatures. Then it is not expected to observe the energy shifts in the in-situ XPS spectra shown in Figure 3b. At least, it is not expected to observe both oxide layer and charge transfer for the same thermal treatment in the classical SMSI effect.

Response: Thank you for this question. We completely agree that the electron transfer

between support and nanoparticles occurs before the migration of oxides from support to the nanoparticle surface at lower temperatures, because the migration was induced by the electron transfer. However, no evidence showed that the electron transfer disappeared after migration. Actually, many works found the electron transfer by XPS co-existed with oxide layer after high temperature treatments in the SMSI process (*Nature Catalysis*, 2021, 4: 418-424; *ACS Catalysis*, 2021, 11: 6081-6090; *Science Advances*, 2017, 3, e1700231).

14) Why do the authors observe Cu 2p XPS spectra with almost the same intensity at Figure 3 if the sample after reduction treatment is covered by an oxide layer? The Cu 2p XPS intensity should decrease significantly in this case.

Response: This is a good question. As we known, the detection depth of X-ray in XPS technology is generally several nanometers (in our test it is about 4-5 nm). If the metal particles were larger than 5 nm, the intensity should significantly decrease. But in our case, the Cu particles were 2-3 nm in size and the oxide layer was about 1 nm in thickness. The total diameter including Cu nanoparticle and oxide layer is 3-4 nm, which is slightly lower than the detection depth of XPS in our test. Therefore, the signal of Cu 2p may not be severely affected by the oxide layer.

15) It is not clear the existence of Cu(0) instead of Cu₂O from both XPS/AES and XAS data. The Cu L₃M_{4,5}M_{4,5} Auger spectrum from Cu(0) presents 4 distinct contributions and that from Cu₂O presents 2 main contributions. See the corresponding figures of the cited classical references (*Surf. Sci. Spectra*, 1993, 2, 55 for Cu₀ and *Surf. Sci. Spectra*, 1998, 5, 257 for Cu₂O). The Auger spectrum enables easy distinction of Cu₀ to Cu₂O

by the shape of the Auger spectrum, which is consistent with the presence of Cu₂O in this work (see Figure S17). Regarding XAS data, the XANES analysis are not shown by the authors for these samples. It is only shown a comparison with the Cu(0) and CuO standards but it is not shown the XANES spectrum for the Cu₂O standard. From the fingerprint existing at XANES of Cu₂O in the literature, I would expect a combination of both Cu(0) and Cu₂O compounds from the XANES analysis. The procedure adopted for fitting the FT is not shown and the fit quality is shown only for 2 cases.

Response: Thanks for the valuable suggestion. We completely agree with the reviewer that the Cu species of Cu(0) and Cu(1) were better to be distinguished by the shape rather than the peak position in Cu LMM Auger spectrum, since the peak position may be affected by supports. Usually, the Auger spectrum for Cu(0) showed one significant peak and three weak peaks, and the one for Cu(1) showed two peaks with similar intensity. From the Auger spectrum in Supplementary Fig. S22, one peak was observed in 250R and 500R samples, and two peaks were detected in ROR sample, indicating that Cu(0) mainly existed in the former and Cu(1) species formed after ROR treatment. This discussion was revised on Page 17 in revised manuscript.

The Cu K-edge X-ray absorption near edge structure (XANES) of different samples was added in Fig. 1i with the Cu(0), Cu₂O and CuO standards. The XANES spectrum analysis by linear combination fitting was conducted and added in Table S1.

Table S1 | Analysis of Cu component. Linear combination fitting results of Cu K XANES data of IM-Cu/TiO₂-500R, SP-Cu/TiO₂-500R, SP-Cu/LaTiO₂-500R and SP-Cu/LaTiO₂-ROR samples.

	Cu (%)	Cu ₂ O (%)	CuO (%)	R factor
IM-Cu/TiO ₂ -500R	0.0±3.8	11.3±4.2	88.5±2.1	0.006
SP-Cu/TiO ₂ -500R	0.0±3.7	6.9±4.1	92.6±2.1	0.006
SP-Cu/LaTiO ₂ -500R	70.1±2.0	17.2±2.2	14.4±1.1	0.002
SP-Cu/LaTiO ₂ -ROR	0.0±3.5	8.7±3.8	90.8±1.9	0.005

16) Figure 3 and S16 are not convincing in the XPS part. The Ti 2p XPS data are shown without the fit of the data. The Cu 2p XPS spectra are not properly fitted since there is a component at around 931.7 eV in the ROR case that is not included in the fit and the components used are not properly identified because there are missing identification. The fit procedure should be described and the fit result should be included. Furthermore, why the XPS spectrum of SP-Cu/LaTiO₂-ROR sample is different than that of SP-Cu/LaTiO₂-500R samples but such difference is not observed in the AES spectra?

Response: Thanks for the good advice.

Firstly, the fitted Ti 2p XPS data were supplemented in revised Supplementary Fig. S21 as below.

For the second question, the Cu 2p XPS spectra of ROR sample was re-processed, and the new fitting results were added in Fig. 3c. The fit procedure for Cu 2p spectra was shown in Fig. S20. The Cu 2p_{3/2} of Cu⁰ and Cu²⁺ species is fixed at the binding energy of 932.3 and 934.5eV, respectively. Cu 2p_{1/2} is identified according to split spin-orbit components of Cu 2p peak with the fixed splitting of the doublet Δ=19.75eV and intensity ratio at 0.508. where the peak at 934.5 eV for Cu 2p_{3/2} can be assigned to Cu²⁺ species. This fitting description was added in the revised manuscript on Page 17.

For the third question, the Cu²⁺ species can be fully reduced in Cu/LaTiO₂-500R,

while those formed during oxidation procedure cannot be fully reduced in the following low temperature reduction step for SP-Cu/LaTiO₂-ROR sample. So, the difference between SP-Cu/LaTiO₂-ROR and Cu/LaTiO₂-500R samples is mainly at Cu²⁺ species, which have much differences in the 2p spectra. However, the signal in the AES spectra has less difference for Cu⁰ and Cu²⁺ species.

Figure S21 | Electron transfer after redox treatments. *In-situ* XPS Ti 2p spectra of SP-Cu/LaTiO₂ sample after different treatments. The fresh sample was reduced at 500°C for 1 h in 99.99% H₂ flow at 0.5 MPa. The XPS spectra were measured when cooling down to room temperature (500R). Subsequently, the sample was oxidized at 400°C for 1 h in 10% O₂/He flow at 0.1 MPa and cooled down to 250°C for another reduction process for 1 h in 99.99% H₂ flow at 0.5 MPa. The XPS spectra were measured again after reduction (ROR).

17) The TiO₂ support may present sample charging effects during XPS measurements. How it was handled by the authors?

Response: Thanks for the good question. In the present work, X-ray photoelectron

spectra (XPS) were recorded on the Thermo Scientific ESCALAB 250Xi spectrometer with a monochromatic Al K α source radiation. The binding energy was calibrated by contamination carbon C 1s peak (284.8 eV) as the reference to eliminate any energy shifts caused by charging effects.

18) The authors should include the Survey XPS spectra showing the absence of contamination.

Response: We appreciated the good advice very much. The Survey XPS spectra was added in the revised Supplementary Fig. S7.

Figure S7 | Analysis of Ti-O-La structure. Survey (a), La 3d (b), O 1s (c) and Ti 2p (d) XPS spectra of SP-Cu/LaTiO₂-fresh sample.

19) Table S2: Cu loading is not the same between the samples, how do the authors compare the catalytic results?

Response: This is really a good question. Considering the variation of exposed active sites with coverage extent, the exposed Cu surface area of tested catalysts were measured and listed, accompanied with Cu loading and dispersion in Supplementary Table S3. The reaction rate was based on the amount of reacted CO₂ per exposed Cu surface area in one hour ($\text{mmol}_{\text{CO}_2} \cdot \text{m}^2_{\text{Cu}}^{-1} \cdot \text{h}^{-1}$). High extent encapsulation is more effective in stabilizing Cu nanoparticles, but will sacrifice more active sites. Although the initial reaction rate of SP-Cu/LaTiO₂ was lower than SP-Cu/TiO₂, high stability of the former made it more competitive. Anyway, SMSI strength should be available to be regulated to find a balance between activity and stability.

20) It was not measured in-situ XPS spectra because, after treating the sample inside the XPS chamber, the XPS spectra were “measured when cooling down to room temperature”. In-situ XPS spectra would be measured at the thermal treatment temperature and with the sample exposed to the gas. By the way, were the XPS spectra measured in vacuum during cooling? What was the pressure?

Response: This is really a good comment. The XPS measurement in the present work is conducted in vacuum after cooling. We agree with the reviewer that *in-situ* XPS spectra should be measured at the thermal treatment temperature and with the sample exposed to the gas. To clearly understand, we revised “*in-situ* XPS” into “*quasi in-situ* XPS” in the revised manuscript.

21) No details are given about the sample preparation for the measurements performed.

Also, no details are given about the procedure adopted to analyze the data obtained, specially XPS and XAS spectra.

Response: We appreciated the good advice. Sample preparation and measurement procedure were in detail described for XPS as “*The catalysts powder was ground and pelletized for measurement. XPS measurements of C 1s, O 1s, Cu 2p, Cu LM2, Ti 2p and La 3d binding energies were conducted at room temperature for the fresh sample (Fresh). Then it was reduced at 500 °C for 1 h in 99.99 % H₂ flow at 0.5 MPa. After cooling to room temperature, the samples were inertly transferred to a glove box connected to the spectrometer and mounted on the sample holder. They were directly transferred from the glove box to the spectrometer chamber without air contact.*” on Page 27 and for XAS as “*Cu metal foil, CuO and Cu₂O were pressed into pellets. All the tested samples were ground into fine powder and uniformly coated on a tape in the air. The time-resolved XAS spectrum was analyzed by linear combination analysis (LCA) with three standard spectra: Cu, Cu₂O and CuO. EXAFS data were processed and analyzed using the IFFEFIT software package⁵³. Artemis was used to fit the experimental data with model structures of metallic Cu (43493) and CuO (16025) from Inorganic Crystal Structure Database (ICSD). Structural parameters included energy shift of the path (ΔE_0), change in the half path length (ΔR), amplitude reduction factor (S_0^2), coordination number (CN) and relative mean-square displacement of the atoms included in path (Debye-Waller factor, σ^2)*” on Page 29 in the revised manuscript.

Reviewer #3 (Remarks to the Author):

It is a very interesting paper proposing a novel method for preparing high stability Cu-based catalysts with SMSI. The paper is easy reading and very well organised. Many modern techniques have been used for the characterisation of the samples while the main conclusions are well supported by the experimental results. In my opinion the manuscript can be published after minor revisions on the following points which need clarification/discussion.

Introduction: Authors should discuss in more detail two aspects, the mechanism of the reconstruction of the metal electronic structure using the sputtering technique and how the reducibility of the support enhances the capability of the metal for hydrogen dissociation. A few lines are written in the Results however discussion is also necessary in Introduction.

Response: Thank you for the good advice. For sputtering (SP) technique, the electronic structure of Cu nanoparticles was reconstructed by the bombardment of high-energy plasma, stabilizing the outmost electron. The reconstruction changed Cu into an electron acceptor, which induced the migration of TiO_x species. For flame spray pyrolysis (FSP) method, the lattice distortion was strengthened by quenching from extremely high temperature, enhancing the activity of lattice oxygen. It could remarkably enhance the reducibility of TiO_2 , and thus compensate the low capability of Cu in dissociating H_2 . These discussion was added in the introduction section on Page 4 of the revised manuscript.

Results: Please explain why La enhances the reducibility of the TiO_2 .

Response: This is really a good question. In general, reduction is necessary in SMSI process to create a sub-stoichiometric state with lower oxygen concentrations on

reducible oxide supports, and induce oxides migration to the surface of metal nanoparticles. However, the reduction temperature of TiO₂ was too high. On one hand, La atoms substituted Ti atoms in the form of Ti–O–La, inducing lattice disorder by the radius disparity between La³⁺ (1.03 Å) and Ti⁴⁺ (0.61 Å), making the lattice oxygen more active. On the other hand, high degree of non-stoichiometry active oxygen can be created during high temperature quenching process in FSP method. These functions improved the reducibility of LaTiO₂ support, making it easier to be induced to immigrate to the metal surface.

Results-Figure S2. Is the term dynamic capsulation process the appropriate one ? Is this an in-situ experiment ? The examined area of the sample (in these 3 images) is always the same ?

Response: Thank you for the good question. The term “dynamic capsulation process” was changed into “dynamic encapsulation process” which is more commonly used. This is really an *in-situ* experiment. The same examined area of the Cu particle was exposed to the electron beam for different time to observe the dynamic encapsulation process.

Results-Figure2a. It is really amazing the stability of the SP-Cu/LaTiO₂ catalyst. It is also very interesting the initial increase of the rate (during the first 100 h). According to the authors this is due to the decrease of the size of the particles. Please give a possible mechanism explaining how a possible metal re-dispersion at 600 oC can lead in metal particle decrease.

Response: This is a good question. The re-dispersion function on metal particles is one

of the classical features of SMSI. During high temperature reduction process, large amount of oxygen vacancies was produced on the reducible support. Meanwhile, the migration of Cu atoms from the Cu nanoparticles to the support will happen during the thermal motion process. These Cu atoms will be captured and settled by the oxygen vacancies under the strong metal-support interaction, inducing the re-dispersion of metal particles. More references and discussions about the re-dispersion function of SMSI were added on Page 12 of the revised manuscript.

Results-page 13 (lines 1-4). Authors give DFT calculations based on a specific mechanism (RWGS reaction pathway). However, other possible steps may be also possible (like CO₂ dissociative adsorption).

Response: This is another excellent comment. We completely agree with the referee. In general, the mechanism of RWGS reaction includes two routes, including COOH-mediated mechanism, and CO₂ dissociative to CO route. However, there is still controversial in many literatures to date. It probably depends on catalyst structure, such as, various types of metal and support, particle size, lattice plane, and many other factors. In consideration of our encapsulated model, CO₂ dissociation difficulty, and most recent viewpoints (Weckhuysen, et al., *Nat. Catal.*, 2018, 1, 127-134; Bobadilla, et al., *ACS Catal.* 2018, 8, 8, 7455–7467; Maestri et al., *J. Phys. Chem. C* 2015, 119, 4959-4966), the direct CO₂ dissociative process probably is more difficult than H-mediated process in the SMSI series catalysts, and thus we prefer to choose COOH-mediated mechanism as the primary route in our DFT calculations. Followed by the excellent suggestion from the referee, we will also investigate and compare another

possible CO₂ dissociative route by designing further more model catalysts in the future work after comprehensive discussion and assessment with our DFT participator.

Do the authors exclude any catalytic contribution of La ?

Response: Thank you for the good question. The SP-Cu/TiO₂ was prepared as a comparison sample. The initial activity of SP-Cu/TiO₂ was higher than that with La, indicating that La has no catalytic contribution to the CO₂ conversion. But La addition will have positive effect on the stability of Cu catalysts.

REVIEWER COMMENTS

Reviewer #2 (Remarks to the Author):

In the revised version of the manuscript, the authors did not show a clear and direct evidence of the existence of the capping layer covering the small Cu nanoparticles (again only for the big ones). It does not mean this affirmation is wrong, perhaps it is valid, but there is no clear evidence for that. This is the key point of the manuscript. Furthermore, most of the analysis present mistakes in the interpretation, lack of rigor in the data analysis and figures hard to evaluate the fit quality. Considering the high profile of Nature Communications journal, I do recommend to reject the manuscript. Some specific points:

- 1) The precise determination of the size of the fresh Cu nanoparticles is very important in this manuscript. Firstly, it is well known that the onset of the SMSI effect depends on the particle size. Secondly, the manuscript discuss about the thermal stability of Cu nanoparticles due to the SMSI effect but the authors are not able to affirm there is absence of sintering before the SMSI occurrence. Then, how do the authors are confident the sintering effect is not faster than the SMSI occurrence and, in fact, the Cu nanoparticles are thermally stable in this case? I understand the difficulty of getting good TEM images but the nanoparticle size can be determined by other techniques as well. Furthermore, if one trust on the initial nanoparticle size estimated by the authors without an histogram (1 nm), it means there is a strong sintering effect before SMSI occurrence (from 1 nm to 2.3 nm there is an increase of 130% in size). Then, the authors cannot affirm the nanoparticles are thermally stable.
- 2) For the histogram of size distribution, 100 particles is not statistically significant. Furthermore, the standard deviation requested is not shown.
- 3) Figure S2 brings additional issues on the discussion. It shows some humps of few nanometers in the edge of the dark region, which should be ascribed to the Cu nanoparticles. However, these nanoparticles do not present any oxide layer and, in this case, the contrast between the oxide layer, the carbon film of the grid, and the Cu nanoparticle surface should be enough to be distinguished. Note that I'm referring to the humps existing in this image and certainly in other regions since the Cu nanoparticles were supported on LaTiO₂. This kind of image of very small humps with the oxide layer is observed in the literature since a long time ago (see, for example, S. Bernal et al, *Cat. Today*, 77, 385 (2003)). Regarding the CO adsorption measurements, it shows the presence of a capping layer, which can be present at the small or big particles. Whether most of the particles are big after thermal treatment, the capping layer is present (as demonstrated by TEM) and the CO adsorption measurements agree with it, but the sintering effect occurred before the claimed thermal stability by the SMSI effect. The authors may affirm that most of the particles are small after thermal treatment, but it is hard to precisely determine the proportion of small/big particles by TEM, which may depend even on the sample preparation method.
- 4) The Cu nanoparticles without the capping layer may present a metallic component from XANES analysis since it is present at the core of the nanoparticle and it turns into the initial question about the size of the Cu nanoparticles. Big particles should have a metallic core, so the existence of metallic Cu does not imply on the existence of the oxide layer at the surface of the nanoparticles. Furthermore, besides authors including a Cu-O scattering path for fitting the FT in the new version of the manuscript, there is no component at that position of FT, as observed in Figure S6. In fact, the region is very similar to the FT of the Cu foil, where no Cu-O scattering path should be present.

7) The FWHM value of a 3d_{5/2} XPS component should be the same than the 3d_{3/2} XPS one. Both 3d_{5/2} and 3d_{3/2} components should give the same information about the chemical components present at the surface. Then the fit shown is meaningless. Furthermore, it is very hard to distinguish 2 components in the XPS spectrum at 3d_{5/2} region due to the very noisy data.

11) I agree about the possibility of re-dispersion of the metal nanoparticles but it is not clear that it is possible to decide between it or Ostwald ripening process in the first 100 h, as claimed by the authors.

12) I'm not convinced about the noise level, the authors should go further than 4 Å in distance in order to affirm about the noise level.

14) The inelastic mean free path, in this case, is around 0.8 nm for the photoelectrons coming from the Cu 2p electronic level, instead of 4-5 nm. Then it is expected to see a change on the Cu 2p XPS intensity in this case.

15) The Auger spectra of Figure S22 are very similar between samples and all of them are consistent with the Cu₂O compound. Considering the noise level, it is very hard to determine the ROR sample presents 2 peaks at 918.5 eV and 916.6 eV and the reduced one presents a single peak at 917.1 eV. The XANES data show differences between samples, besides probing a different depth than the Auger data. However, the fit of the XANES data are not shown. The requested fit of the FT, again, was not shown for all cases.

16) It is hard to evaluate the fit quality of the XPS spectra in Figure S21 because the experimental points are not visible and the yellow components overlap the data. Even so, there is a shoulder in small binding energy for the 500R sample that is not fitted. The fit procedure does not consider constraints in the FWHM of a given component, then the fit results may be incorrect. Finally, the last question is replied considering the similarity between Cu₀ and Cu⁺² Auger spectra, which is not true as can be observed in the literature (Surf. Sci. Spectra, 1993, 2, 55 for Cu₀ and Surf. Sci. Spectra, 1998, 5, 262 for CuO). Then the XPS analysis remains with several issues not properly fixed by the authors.

17) This procedure eliminates the charging effect in the binding energy position but it does not correct for peak asymmetry effects, which are common to occur for metal oxides. How do the authors are confident that such effects are not influencing on the results presented?

18) There are non-identified peaks in low binding energy (less than 200 eV). Some of them are photoemission peaks from Cu, O or Ti but others are impurities in the sample.

21) There are still important details missing in the experimental procedure (XPS – energy pass, energy resolution, dwell time, pressure at the chamber, heating rate, etc – XAS – mass of sample, detection mode, step size, time/point, etc). Furthermore, the procedure of XAS data reduction is not described and the fit procedure of EXAFS has no constraint included, which means there is a great probability of existing mistakes in the data obtained from the EXAFS analysis.

Reviewer #3 (Remarks to the Author):

In my opinion the manuscript has been thoroughly revised according to the reviewers' comments and could be published in its current form.

2nd Point-by-point responses to the comments

Manuscript ID: NCOMMS-21-26046A; NCOMMS-21-26046-T

Title: Ultra-high Thermal Stability of Sputtering Reconstructed Cu-based Catalysts

Dear reviewers,

We are very grateful to you for the very valuable and insightful comments on our manuscript. It is a very good opportunity to help us improve the whole quality of the manuscript. We appreciate two copies of positive publication comments from reviewer 1 and 3, and also very excited to carefully discuss critical comments with reviewer 2. In the 2nd revised version, all of these comments are carefully answered and additional results are provided. We believe that the revised manuscript provides a clearer and in-depth insight on ultra-high thermal stability of sputtering reconstructed Cu-based catalysts, which will be meaningful to wide catalysis and material researchers. The following descriptions are the responses to these excellent comments one by one.

Thanks again for your reviewing.

Kind regards,

Prof. Jian Sun, Prof. Noritatsu Tsubaki & Prof. Yuefeng Liu

Author Response to Reviewer #2

In the revised version of the manuscript, the authors did not show a clear and direct evidence of the existence of the capping layer covering the small Cu nanoparticles (again only for the big ones). It does not mean this affirmation is wrong, perhaps it is valid, but there is no clear evidence for that. This is the key point of the manuscript. Furthermore, most of the analysis present mistakes in the interpretation, lack of rigor in the data analysis and figures hard to evaluate the fit quality. Considering the high profile of Nature Communications journal, I do recommend to reject the manuscript. Some specific points:

1) The precise determination of the size of the fresh Cu nanoparticles is very important in this manuscript. Firstly, it is well known that the onset of the SMSI effect depends on the particle size. Secondly, the manuscript discuss about the thermal stability of Cu nanoparticles due to the SMSI effect but the authors are not able to affirm there is absence of sintering before the SMSI occurrence. Then, how do the authors are confident the sintering effect is not faster than the SMSI occurrence and, in fact, the Cu nanoparticles are thermally stable in this case? I understand the difficulty of getting good TEM images but the nanoparticle size can be determined by other techniques as well. Furthermore, if one trust on the initial nanoparticle size estimated by the authors without an histogram (1 nm), it means there is a strong sintering effect before SMSI occurrence (from 1 nm to 2.3 nm there is an increase of 130% in size). Then, the authors cannot affirm the nanoparticles are thermally stable.

Response: The authors appreciate the critical comments raised by the reviewer. Before answer above questions, we would like to highlight the main concepts and developments in our work. It is well known that Cu nanoparticles are easily sintered at a high temperature (usually above 300°C), that is the main reason we devote in the establishment of SMSI and improve the Cu thermal stability to broaden their applications in high temperature reactions. Therefore, we are unable to suppress the

ultra-small particles sintering before SMSI formation. We confirmed the Cu nanoparticles are extremely **stable after SMSI formation** from the stability tests and a great number of characterizations even at reaction temperature as high as 800 °C under H₂-containing reaction conditions. We are confident such achievement may bring the great significant in the field of the catalysis. But we never say they can be stable before SMSI (actually they cannot as have mentioned by the reviewer).

The MAIN contribution of the present work is to supply a strategy to establish SMSI on Cu nanoparticles and obtain a thermally stable catalyst during high temperature reaction after SMSI formation. The main conclusion and innovation of this work will not be affected by the fact that Cu nanoparticles increase from 1 to 2.3 nm in size before SMSI occurrence. Actually, this is a very common phenomenon in other SMSI-related works. Honestly, Cu particles in size of 2.3 nm before SMSI are small enough to perform a high activity in catalytic reactions, and the key point we should concern is to suppress these small particles from sintering during reaction rather than where they come from.

2) For the histogram of size distribution, 100 particles is not statistically significant. Furthermore, the standard deviation requested is not shown.

Response: Thanks very much for the reviewer's comment. We fully agree with the reviewer's comment that the number of 100 particles is not statistically significant for particle size distribution. Due to the fact that the Cu NPs in SP-Cu/LaTiO₂-500R sample are covered by oxide layer, the particles are hard to identify by microscope. In this case,

we tried to calculate up to 100 particles for size distribution. But the other samples are calculated using more than 200 particles for the histogram. Although there have few big nanoparticles of 5-10 nm, the most of the Cu nanoparticles are well fixed in the particle size range according to the STEM/TEM images of the samples. The standard deviation has been added in all histogram of size distribution. Please see the revised Figure S1, S3 and S12.

Figure S1 | Distribution of Cu nanoparticles in different samples. TEM images and particles size distributions of (a) IM-Cu/TiO₂-500R, (b) SP-Cu/TiO₂-500R and (c) SP-Cu/LaTiO₂-ROR. The average particle size of Cu in different samples are 2.3, 3.0, 2.8 nm, respectively.

Figure S3 | Identification of Cu nanoparticles. a, Atomic resolution annular dark-field scanning transmission electron microscope (ADF-STEM) images of SP-Cu/LaTiO₂-500R. The average Cu particle size is 2.3 nm. b, ADF-STEM image and c, element mapping analysis of SP-Cu/LaTiO₂-

500R sample. Cu nanoparticles were well dispersed on LaTiO₂ support. La was also well dispersed in TiO₂ without forming large particles.

Figure S12 | TEM image and distribution of Cu nanoparticles. HRTEM images of (a) IM-Cu/TiO₂ catalyst after reduction at 500°C for 1h (IM-Cu/TiO₂-500R) and (b) IM-Cu/TiO₂ catalyst after RWGS reaction at 600°C for 50 h (IM-Cu/TiO₂-50h), as well as (c) particle size distributions of reduced and used catalyst. The particle sizes of Cu in reduced and used samples were 2.3 ± 0.6 and 4.7 ± 1.2 nm, respectively.

3) Figure S2 brings additional issues on the discussion. It shows some humps of few nanometers in the edge of the dark region, which should be ascribed to the Cu nanoparticles. However, these nanoparticles do not present any oxide layer and, in this case, the contrast between the oxide layer, the carbon film of the grid, and the Cu nanoparticle surface should be enough to be distinguished. Note that I'm referring to the humps existing in this image and certainly in other regions since the Cu nanoparticles were supported on LaTiO₂. This kind of image of very small humps with the oxide layer is observed in the literature since a long time ago (see, for example, S. Bernal et al, *Cat. Today*, 77, 385 (2003)). Regarding the CO adsorption measurements, it shows the presence of a capping layer, which can be present at the small or big particles. Whether most of the particles are big after thermal treatment, the capping layer is present (as demonstrated by TEM) and the CO adsorption measurements agree

with it, but the sintering effect occurred before the claimed thermal stability by the SMSI effect. The authors may affirm that most of the particles are small after thermal treatment, but it is hard to precisely determine the proportion of small/big particles by TEM, which may depend even on the sample preparation method.

Response: Thanks for the valuable comments. This is really a very detailed question. There are some following opinions from authors on the attribution of Cu nanoparticles in TEM.

First of all, after discussion with several TEM operator and analysis experts, the humps could not be directly identified to Cu particles since no lattice fringes can be observed and no contrast difference of the humps can be distinguished from the support. Herein, we supply more HRTEM images for comparison. The following figure showed TEM and STEM images for the same part of SP-Cu/LaTiO₂-fresh sample in a and b, respectively. There is a hump marked by red circle on the surface of support, but it didn't show in the STEM images as metals did. Therefore, this hump position cannot be directly assigned to Cu metal.

Fig. R1 TEM (a) and STEM (b) images for the same part of SP-Cu/LaTiO₂-fresh sample

Secondly, the identification method of humps in the above literature is probably not proper to be used in the TEM image in Fig. S2. The humps in the literature were supported on a flat surface, for example $\text{TiO}_2(111)$, so the humps can be clearly considered as different species from the support. But the surface of TiO_2 particles in our case is not smooth enough to exclude the existence of TiO_2 humps. More importantly, the metals in the literature were noble metals with higher crystallinity. They were also identified by the contrast differences and lattice fringes but not only by the shapes. So it is unreasonable to identify Cu particles based on their shapes.

Thirdly, if most of the Cu particles are big after thermal treatment, the CO adsorption will decrease after thermal treatment but cannot increase again after ROR treatment, which is inconsistent with the characterization results in Fig.3a. Moreover, if most of the Cu particles are big (above 10 nm as observed in TEM), the activity will not be stable at a high level, because serious deactivation happened when Cu particles increased to 4.7 nm in 50h reaction for IM-Cu/ TiO_2 sample. So this can be excluded.

4) The Cu nanoparticles without the capping layer may present a metallic component from XANES analysis since it is present at the core of the nanoparticle and it turns into the initial question about the size of the Cu nanoparticles. Big particles should have a metallic core, so the existence of metallic Cu does not imply on the existence of the oxide layer at the surface of the nanoparticles. Furthermore, besides authors including a Cu-O scattering path for fitting the FT in the new version of the manuscript, there is no component at that position of FT, as observed in Figure S6. In fact, the region is very similar to the FT of the Cu foil, where no Cu-O scattering path should be present.

Response: If big particles are oxidized on the surface with a metallic core inside as the

reviewer said, the oxidized proportion should increase with decreasing Cu size. If it is true, the XAS signal of SP-Cu/LaTiO₂-500R would exhibit obvious Cu-O scattering since it gives all of the coordination environment of Cu atoms. Otherwise, the Cu particles should be as big as Cu foil, considering that the XAS result of SP-Cu/LaTiO₂-500R showed similar signal with Cu foil. Apparently, it is unreasonable.

Moreover, the metallic Cu signal in XAS cannot be derived from the core of big particles, because the contrastive SP-Cu/TiO₂ sample was completely oxidized, considering that the SP-Cu/TiO₂ samples were made by the same method and consisted of big particles as well. As we emphasized in the manuscript, only few large particles were found to facilitate the observation of encapsulation layer. Most of the particles were in small size with an average of 2.3 nm, which was fully supported by the STEM images of SP-Cu/LaTiO₂-500R sample and the TEM of SP-Cu/LaTiO₂-ROR sample.

For the second question, although the FT of SP-Cu/LaTiO₂-500R is very similar to the FT of the Cu foil, Cu-O scattering path should still be included for accurately and fairly fitting, because the fitting method needs to be consistent for all samples. In principle, if it is possible for Cu-O scattering to exist like other samples, we cannot identify whether there is or not Cu-O scattering path by subjective feelings, but should see what the fitting results show.

7) The FWHM value of a 3d_{5/2} XPS component should be the same than the 3d_{3/2} XPS one. Both 3d_{5/2} and 3d_{3/2} components should give the same information about the chemical components present at the surface. Then the fit shown is meaningless.

Furthermore, it is very hard to distinguish 2 components in the XPS spectrum at 3d5/2 region due to the very noise data.

Response: We completely agree with the reviewer on the fact that 3d5/2 and 3d3/2 components should give the same information about the chemical components. In this revision, The La 3d XPS spectra was re-fitted to keep the same component for 3d5/2 and 3d3/2. Although much noise existed due to low content of La (3.8wt%), it can be obviously seen that the peak at 834.8 eV split into two peaks as shown in Fig. S8b in the revised manuscript and the original data for reviewer only.

Figure S8 | Analysis of Ti-O-La structure. Survey (a), La 3d (b), O 1s (c) and Ti 2p (d) XPS spectra of SP-Cu/LaTiO₂-fresh sample.

Fig. R2 The original XPS spectra of La 3d without fitting for reviewer only.

11) I agree about the possibility of re-dispersion of the metal nanoparticles but it is not clear that it is possible to decide between it or Ostwald ripening process in the first 100 h, as claimed by the authors.

Response: We totally agree with the reviewer on this point. Both of the two processes might exist in the first 100h. The catalyst became stable after reaching the balance state in the synergistic effect of these two processes. We also believe the confirmation of re-dispersion or Ostwald ripening in the SMSI catalyst is of great importance in future. The reviewer points out a new potential research direction in heterogeneous catalysis. We will further investigate that in the coming new research. Thanks again.

12) I'm not convinced about the noise level, the authors should go further than 4 Å in distance in order to affirm about the noise level.

Response: The distance was prolonged to 2.8 nm in the revised Fig. S5, where the noise level was only about 2 counts.

Figure S5 | Identification of encapsulation. ADF-STEM image (a) and line scanning elemental analysis (b) of SP-Cu/LaTiO₂-500R sample. The line profile was collected along the yellow arrow in the image with a noise level of 2 counts. Ti species can be detected on the half of Cu surface, indicating that an about 0.8 nm Cu nanoparticle was encapsulated by TiO_x species as the profile described in the inset image.

14) The inelastic mean free path, in this case, is around 0.8 nm for the photoelectrons coming from the Cu 2p electronic level, instead of 4-5 nm. Then it is expected to see a change on the Cu 2p XPS intensity in this case.

Response: This is a very good question. The detection depth of X-ray is about 3 times of the inelastic mean free path, i.e., about 2-3 nm in this case, which is nearly the level of the Cu nanoparticle size. Some literatures reported the XPS detection depth was 4-5 nm (*Applied Surface Science*, 2013, 272, 82–87; *Surf. Interface Anal.* 2002, 33, 869–878). Moreover, the oxides layer may encapsulate half of Cu nanoparticles, as can be seen in EDS analysis in Fig. S5. So it is reasonable that the change derived from the effect of encapsulation layer was not clearly detected. The existence of encapsulation layer was also proved by many other characterizations, which is not contradictory to the absence of signal changes.

15) The Auger spectra of Figure S22 are very similar between samples and all of them are consistent with the Cu₂O compound. Considering the noise level, it is very hard to determine the ROR sample presents 2 peaks at 918.5 eV and 916.6 eV and the reduced one present a single peak at 917.1 eV. The XANES data show difference between samples, besides probing a different depth than the Auger data. However, the fit of the XANES data are not shown. The requested fit of the FT, again, was not shown for all cases.

Response: We agree with the reviewer that it is very hard to determine the Cu species considering the noise level in Auger spectra of Figure S22. In consideration of the limited contribution to the main concept, this figure is deleted to avoid misleading. Anyway, the Cu valence states from Auger spectra cannot compare with those from XANES data. It should be noted that the samples were *in-situ* treated before XPS measurement, but they exposed to the air for a long time before XAS experiment. The fit of XANES and EXAFS data are shown in the revised manuscript in Fig. S6 and Fig. S24.

Figure S6 | Analysis of Cu component. Linear combination fitting results of Cu K XANES data of IM-Cu/TiO₂-500R, SP-Cu/TiO₂-500R, SP-Cu/LaTiO₂-500R and SP-Cu/LaTiO₂-ROR samples after exposing to the air at room temperature for a long time. Details can be found in Table S1.

Figure S24 | EXAFS fitting results. Cu K-edge fitting results of fresh SP-Cu/LaTiO₂, SP-Cu/LaTiO₂-250R, SP-Cu/LaTiO₂-500R, SP-Cu/LaTiO₂-ROR and SP-Cu/LaTiO₂-100h. Details of EXAFS fitting can be found in Table S5.

16) It is hard to evaluate the fit quality of the XPS spectra in Figure S21 because the experimental points are not visible and the yellow components overlap the data. Even so, there is a shoulder in small binding energy for the 500R sample that is not fitted. The fit procedure does not consider constrains in the FWHM of a given component, then the fit results may be incorrect. Finally, the last question is replied considering the similarity between Cu⁰ and Cu⁺² Auger spectra, which is not true as can be observed in the literature (Surf. Sci. Spectra, 1993, 2, 55 for Cu⁰ and Surf. Sci. Spectra, 1998, 5, 262 for CuO). Then the XPS analysis remains with several issues not properly fixed by the authors.

Response: We appreciate to the useful suggestion. The experimental points were thickened in the revised figure. The XPS spectra is re-fitted, considering the constrains in the FWHM. The shoulder in small binding energy for the 500R sample was added. The revised figure was shown in Fig. S22 in the revised manuscript.

Figure S22 | Electron transfer after different treatments.

17) This procedure eliminates the charging effect in the binding energy position but it does not correct for peak asymmetry effects, which are common to occur for metal

oxides. How do the authors are confident that such effects are not influencing on the results presented?

Response: Thanks for the useful comment. Surface charging effects occurring in an insulant sample can be avoided using an electron flood gun. The description was added in the revised manuscript.

18) There are non identified peaks in low binding energy (less than 200 eV). Some of them are photoemission peaks from Cu, O or Ti but others are impurities in the sample.

Response: Those peaks are identified and added in Fig. S8a.

21) There are still important details missing in the experimental procedure (XPS – energy pass, energy resolution, dwell time, pressure at the chamber, heating rate, etc – XAS – mass of sample, detection mode, step size, time/point, etc). Furthermore, the procedure of XAS data reduction is not described and the fit procedure of EXAFS has no constrain included, which means there is a great probability of existing mistakes in the data obtained from the EXAFS analysis.

Response: We appreciate this valuable suggestion again. We have further supplied the detailed information as mentioned by the reviewer.

Quasi *in-situ* X-ray photoelectron spectra (XPS) were recorded on the Thermo Scientific ESCALAB 250Xi spectrometer with a monochromatic Al K α source radiation and a spot size of 500 μm in diameter. It operated with an analyzer in constant analyzer energy (CAE) mode with a pass energy and energy step size of 100 eV and 1 eV for survey, respectively. They were 20 eV and 0.05 eV for high resolution spectra. Surface charging effects occurring in an isolant samples were avoided using an electron flood gun. The XPS measurement was conducted under room temperature. XPS

measurements of C 1s, O 1s, Cu 2p, Cu LM2, Ti 2p and La 3d binding energies were conducted at room temperature and atmosphere pressure for the fresh sample (Fresh). Then it was reduced at 500 °C for 1 h in 99.99 % H₂ flow at 0.5 MPa with a heating rate of 10°C/min. XAS spectra were recorded in transmission mode with a step size of 0.5 eV. EXAFS spectra were recorded for 7 minutes. The mass of sample was about 10mg. These were added in the experimental section.

The parameters of EXAFS fitting were shown in Table S2. k range: 3-12Å, spline range: 0-12Å, k-weight=3, distance range: 1-2.8Å. S₀² was fixed at 0.943, and E₀ was refined as a global fit parameter for each sample.

REVIEWERS' COMMENTS

Reviewer #2 (Remarks to the Author):

The main issues were addressed by the authors, then I do recommend publication.

3rd Point-by-point responses to the comments

Manuscript ID: NCOMMS-21-26046A; NCOMMS-21-26046-T

Title: Ultra-high Thermal Stability of Sputtering Reconstructed Cu-based Catalysts

Reviewer #2 (Remarks to the Author):

The main issues were addressed by the authors, then I do recommend publication.

Response: We appreciate the reviewer for the approving of our revision.